# Graph Self-supervised Learning with Accurate Discrepancy Learning

**Dongki Kim**[1,*], **Jinheon Baek**[1,*], **Sung Ju Hwang**[1,2]
KAIST[1], AITRICS[2], South Korea
cleverki@kaist.ac.kr, jinheon.baek@kaist.ac.kr, sjhwang82@kaist.ac.kr

## Abstract

Self-supervised learning of graph neural networks (GNNs) aims to learn an accurate representation of the graphs in an unsupervised manner, to obtain transferable representations of them for diverse downstream tasks. Predictive learning and contrastive learning are the two most prevalent approaches for graph self-supervised learning. However, they have their own drawbacks. While the predictive learning methods can learn the contextual relationships between neighboring nodes and edges, they cannot learn global graph-level similarities. Contrastive learning, while it can learn global graph-level similarities, its objective to maximize the similarity between two differently perturbed graphs may result in representations that cannot discriminate two similar graphs with different properties. To tackle such limitations, we propose a framework that aims to learn the exact discrepancy between the original and the perturbed graphs, coined as *Discrepancy-based Self-supervised LeArning* (D-SLA). Specifically, we create multiple perturbations of the given graph with varying degrees of similarity, and train the model to predict whether each graph is the original graph or the perturbed one. Moreover, we further aim to accurately capture the amount of discrepancy for each perturbed graph using the graph edit distance. We validate our D-SLA on various graph-related downstream tasks, including molecular property prediction, protein function prediction, and link prediction tasks, on which ours largely outperforms relevant baselines[1].

## 1 Introduction

A graph, consisting of nodes and edges, is a data structure that defines a relationship among objects. Recently, graph neural networks (GNNs) [15, 8, 31, 38], which aim to represent this structure with neural networks, have achieved great successes in modeling real-world graphs such as social networks [5], knowledge graphs [1], biological networks [20], and molecular graphs [36]. However, it is extremely costly and time-consuming to annotate every label of the graphs. For example, labeling the properties of molecular graphs requires time-consuming laboratory experiments [11].

Recently, various self-supervised learning methods for GNNs [11, 45, 12, 28] have been studied to overcome this issue of the lack of labeled graph data. The existing self-supervised learning methods can be classified into two categories: predictive learning and contrastive learning. Predictive learning methods [11, 22, 14] aim to learn representations by predicting contexts of a graph (Figure 1, (a-1)). However, predictive learning schemes are limited as they consider only the subgraphical semantics. Contrastive learning [32, 27, 42, 46, 43, 37], which is a popular self-supervised learning approach in the computer vision field [24, 35, 4, 9], can learn the global semantics by maximizing the similarity between instances (images) perturbed by semantic-invariant perturbations (e.g. color jittering or flipping). Graph contrastive learning methods also use a similar strategy, generating positive examples

---

*Equal Contribution.
[1]Code is available at https://github.com/DongkiKim95/D-SLA

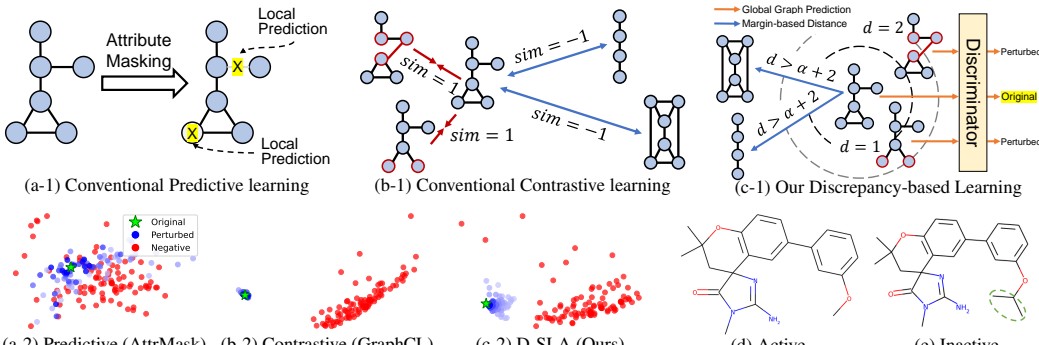

Figure 1: **(a) Conventional predictive learning** that aims to predict local attributes by masking them. **(b) Conventional contrastive learning** that could maximize the similarity of dissimilar graphs perturbed from original graphs. **(c) Our discrepancy-based learning** that discriminates an original graph from perturbed graphs by capturing global semantics of them unlike predictive learning, but also that reflects the amount of discrepancy across original, perturbed, and different graphs, unlike contrastive learning. **(a,b,c-2) Graph embedding visualization** trained by each self-supervised learning scheme. **(d),(e) Motivation:** A pitfall of contrastive learning methods, which assumes two similar graphs as the same, despite of their different properties.

with edge perturbation, attribute masking, and subgraph sampling, and then maximizing the similarity between two differently perturbed graphs from the original graph (See Figure 1, (b-1)).

However, is it always correct to assume that the perturbed graphs by such perturbations will preserve the semantics of the original graph? In computer vision, there exists a set of well-defined perturbations, that does not change the human perception of the given image. However, in contrast to images that reside in a continuous domain, graphs are discrete data structures by nature, and thus their properties could become completely different even with slight perturbations. For example, two molecules in Figure 1 (d) and (e) show that, although they have a highly correlated structure, one molecular graph could be actively bound to a set of inhibitors of human $\beta$-secretase 1 [34], whereas the other is not. In this case, existing contrastive learning with perturbations may lead the representations for two semantically *dissimilar* graphs to collapse into the one representation (See Figure 1 (b-2)).

To tackle this problem, we propose a novel self-supervised learning method which aims to learn the *discrepancy* between graphs, which we refer to as **D**iscrepancy-based **S**elf-supervised **Le**A**rning (D-SLA). Specifically, we first perturb the given graph similarly as with contrastive learning schemes, but instead of maximizing the similarities across the perturbed graphs as done with contrastive learning, we aim to learn their discrepancies. To this end, we first introduce a discriminator which learns to discriminate real graphs from the perturbed ones (See Discriminator in Figure 1, (c-1)). This allows the model to learn small differences that may largely impact the global property of the graph, as illustrated by the example in Figure 1 (d) and (e). However, simply knowing that two graphs are different is insufficient, and we would want to learn the exact amount of discrepancy between them. Notably, with our perturbation scheme, the exact discrepancy can be trivially obtained since the graph edit distance [23] is automatically derived as the number of edges we perturb for a given graph. Then, we enforce the embedding-level difference between original and perturbed graphs to be proportional to the graph edit distance (See dotted circles in Figure 1 (c-1)). Finally, to discriminate the original and perturbed graphs from the completely different graphs, we enforce the relative distance of the latter to be larger than the former, by a large margin (See blue arrows in Figure 1, (c-1)).

This will enable our D-SLA to capture both small and large topological differences (See Figure 1 (c-2)). Thus, our framework can enjoy the best of both worlds for graph self-supervised learning: predictive and contrastive learning. The experimental results show that our model significantly outperforms baselines, especially on the datasets where graphs with different properties have highly correlated structures, demonstrating the effectiveness of ours. Our main contributions are as follows:

- We propose a novel graph self-supervised learning framework with a completely opposite objective from contrastive learning, which aims to learn to differentiate a graph and its perturbed ones using a discriminator, as even slight perturbations could lead to completely different properties for graphs.
- Utilizing the graph edit distance that is obtained for perturbed graphs at no cost, we propose a novel objective to preserve the exact amount of discrepancy between graphs in the representation space.
- We validate our D-SLA by pre-training and fine-tuning it on various benchmarks of chemical, biology, and social domains, on which it significantly outperforms baselines.

## 2 Related Work

We now briefly review the existing works on graph neural networks (GNNs), and self-supervised learning methods for GNNs including predictive and contrastive learning.

### 2.1 Graph Neural Networks

Most existing graph neural networks (GNNs) could be formulated under the message passing scheme [7], which represents each node by firstly aggregating the features from its neighbors, and then combining the aggregated message with its own node representation. Different variants of update and aggregation functions have been studied. To mention a few, Graph Convolutional Network (GCN) [15] generalizes the convolution operation in a spectral domain of graphs, approximated by the mean aggregation. GraphSAGE [8] concatenates the representations of neighbors with its own representation when updating the node representation. Graph Attention Network (GAT) [31] considers the relative importance among neighboring nodes for neighborhood aggregation, which helps identify relevant neighbors for the given task. Graph Isomorphism Network (GIN) [38] uses the sum aggregation, allowing the model to distinguish two different graphs as powerfully as the Weisfeiler-Lehman (WL) test [17]. While GNNs have achieved impressive results on various graph-related tasks, the trained representations from one dataset are usually not transferable to different downstream tasks. Moreover, labels of such task-oriented data are often scarce, especially in scientific domains (e.g., chemistry and biology) [11]. Thus, in this work, we aim at obtaining transferable representations of graph-structured data with self-supervised learning, without using any labels. Self-supervised learning for GNNs can be broadly classified into two categories: predictive learning and contrastive learning, which we will briefly introduce in the following paragraphs.

### 2.2 Predictive Learning for Graph Self-supervised Learning

Predictive learning aims to learn contextual relationships by predicting subgraphical features, for example, nodes, edges, and subgraphs. Specifically, Hu et al. [11] propose to predict the attributes of masked nodes. Also, Hwang et al. [13] and Kim and Oh [14] propose to predict the presence of an edge or a path with the link prediction scheme. Furthermore, Hu et al. [12] and Rong et al. [22] propose to predict the generative sequence, contextual properties, and motifs of the given graphs. However, the predictive learning methods are limited in that they may not well capture the global structures and/or semantics of graphs.

### 2.3 Contrastive Learning for Graph Self-supervised Learning

The limitations of predictive learning gave rise to contrastive learning methods, which aim to capture global graph-level information. Early contrastive learning methods for GNNs aim to learn the similarity between the entire graph and its substructure, to obtain the representations for them without using any perturbations [32, 27]. Despite their successes, to further expand the embedding space of a model, contrastive learning methods have been actively studied with perturbation methods such as attribute masking, edge perturbation, and subgraph sampling [42, 46, 33, 43]. Recently proposed adversarial methods further generate positive examples either by adaptively removing the edges [29] or by adjusting the attributes [40]. Further, Xu et al. [39] aim to cluster graph representations via contrastive learning, and Thakoor et al. [30] propose a memory-efficient method for large-scale graph representation learning. However, while contrastive learning allows learning the global semantics with perturbed graphs, there exists a potential risk where the model can consider two dissimilar graphs as the same (See Figure 1 (b), (d), and (e)), due to the discrete nature of graphs. Contrarily, our novel pretext task of predicting the correct graph from multiple choices, which includes both slightly perturbed graphs and strongly perturbed graphs, allows the model to learn the correct relationships among local elements, but also learn the global graph-level differences.

## 3 Method

In this section, we introduce our novel graph self-supervised learning framework, **D**iscrepancy-based **S**elf-supervised **LeA**rning (*D-SLA*), which is illustrated in Figure 2. Specifically, we first recount the notion of graph neural networks and existing graph self-supervised learning in section 3.1. Then, we introduce each component of our D-SLA in Section 3.2, 3.3, and 3.4, respectively. After that, we finally describe the overall framework that combines all three components in Section 3.5.

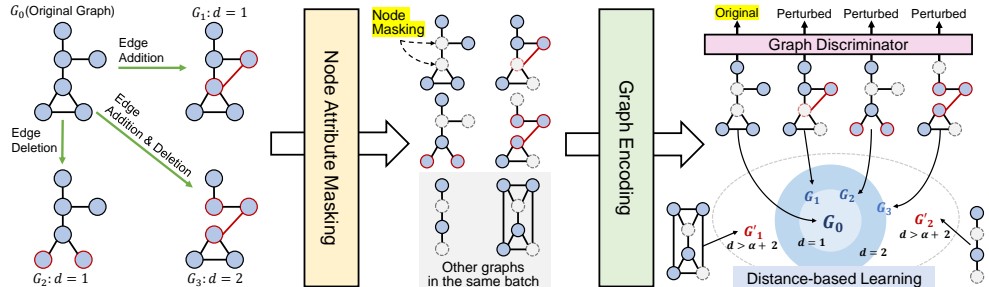

Figure 2: **Illustration of our overall framework (D-SLA).** We perturb the original graph by adding or deleting edges, where the discrepancy between original and perturbed graphs is defined by graph edit distance [23]. Then, we mask the node attributes to make it more difficult to distinguish original and perturbed graphs. After that, we learn GNNs to distinguish perturbed graphs from original ones, but also accurately discriminate the original, perturbed, and other graphs on the embedding space with their distances.

## 3.1 Preliminaries

**Graph Neural Networks** A graph $\mathcal{G}$ can be defined by four components: $\mathcal{G} = (\mathcal{V}, \mathcal{E}, \mathcal{X}_\mathcal{V}, \mathcal{X}_\mathcal{E})$, where $\mathcal{V}$ is the set of nodes, $\mathcal{E}$ is the set of edges, $\mathcal{X}_\mathcal{V} \in \mathbb{R}^{|\mathcal{V}| \times d}$ is the matrix of node features, and $\mathcal{X}_\mathcal{E} \in \mathbb{R}^{|\mathcal{E}| \times d'}$ is the matrix of edge features, where $d$ and $d'$ are the dimensionalities of node and edge attributes, respectively. Let $h_v$ be a representation of a node $v$, and $h_\mathcal{G}$ be a representation of a graph $\mathcal{G}$. Then, the goal of graph neural networks (GNNs) is to represent nodes of the given graph by leveraging its topological structure. To be specific, GNNs learn node representations (i.e., $h_v$) by iteratively aggregating messages from their neighbors in a layer-wise manner, which is usually referred to as message-passing [7] and formally defined as follows:

$$h_v^{(l+1)} = f_U^{(l)} \left( h_v^{(l)}, f_A^{(l)} \left( \left\{ h_u^{(l)} : \forall u \in \mathcal{N}(v) \right\} \right) \right), \tag{1}$$

where $f_U^{(l)}$ denotes an update function that updates a representation of the given node along with its neighbors' representations, $f_A^{(l)}$ denotes an aggregate function that aggregates messages from the node's neighbors, $\mathcal{N}(v)$ denotes a set of neighbors of the node $v$, and $l$ denotes a $l$-th layer of GNNs.

To further obtain the representation for an entire graph, we usually summarize all node representations from Equation 1 into a single embedding vector with a permutation-invariant function $f_R$ as follows:

$$h_\mathcal{G} = f_R \left( \left\{ h_v^{(L)} : \forall v \in \mathcal{V} \right\} \right). \tag{2}$$

While a natural choice of $f_R$ is to use simple operations, such as mean or sum, various graph representation learning methods have been recently studied to accurately capture the entire graph information, including node clustering based [41, 2], and node drop based methods [6, 16].

**Contrastive Learning for Self-supervised Learning of GNNs** We now describe why contrastive learning fails to distinguish two topologically similar graphs yet having completely different properties. In particular, contrastive learning aims to increase the similarity between positive pairs of graphs while increasing the dissimilarity between negative pairs [42, 46, 43]. Threat, to define positive pairs, they perturb the original graph by manipulating edges, masking attributes, and sampling subgraphs, and then consider the perturbed graphs as the same as the original graph. On the other hand, they consider other graphs in the same batch as dissimilar. Formally, for the perturbed graphs $\mathcal{G}_i$ and $\mathcal{G}_j$ from the original graph $\mathcal{G}_0$, the objective of contrastive learning is defined as follows:

$$\mathcal{L}_{CL} = -\log \frac{f_{\text{sim}}(h_{\mathcal{G}_i}, h_{\mathcal{G}_j})}{\sum_{\mathcal{G}', \, \mathcal{G}' \neq \mathcal{G}_0} f_{\text{sim}}(h_{\mathcal{G}_i}, h_{\mathcal{G}'})}, \tag{3}$$

where $\mathcal{G}'$ is the other graph in the same batch with the graph $\mathcal{G}_0$, which is also referred to as a *negative graph*. Therefore, $\mathcal{G}_i$ and $\mathcal{G}_j$ are a positive pair, whereas, $\mathcal{G}_i$ and $\mathcal{G}'$ are a negative pair. $f_{\text{sim}}$ denotes a similarity function between two graphs, for example, $L_2$ distance or cosine similarity. By minimizing the objective in Equation 3, existing contrastive learning methods closely embed two perturbed graphs $\mathcal{G}_i$ and $\mathcal{G}_j$: $\mathcal{G}_i \sim \mathcal{G}_j$. However, as the perturbations may not preserve the properties of the given graph due to the discrete nature of graph-structured data, they indeed should not be similar: $\mathcal{G}_i \not\sim \mathcal{G}_j$ (See Figure 1 (d) and (e)). To this end, we propose a new objective that can differentiate between the original graph and its perturbations, which we describe in the next subsection.

## 3.2 Discrepancy Learning with Graph Discrimination

We now introduce our novel graph self-supervised learning objective to preserve the discrepancy between two graphs in the learned representation space, for which we then describe the perturbation scheme to generate augmented graphs from the original graph.

**Discriminating Original Graphs from Perturbed Graphs** Contrarily to contrastive learning that trains perturbed graphs to be similar, our goal is to distinguish the original graph from perturbed graphs, by predicting the original graph among original and perturbed graphs, which we term as the original graph discrimination (See Graph Discriminator in Figure 2). Let $\mathcal{G}_0$ be an original graph, and $\mathcal{G}_i$ be a perturbed graph with $i \geq 1$. Then objective of our discrimination scheme is as follows:

$$\mathcal{L}_{GD} = -\log\left(\frac{e^{S_0}}{e^{S_0} + \sum_{i\geq 1} e^{S_i}}\right) \text{ with } S = f_S(h_{\mathcal{G}}), \tag{4}$$

where $S_k$ is a score of the graph $\mathcal{G}_k$, obtained from a learnable score network $f_S$ of the discriminator, i.e., $f_S : h_{\mathcal{G}} \mapsto S \in \mathbb{R}$. By training to discriminate the original graph from the perturbed graphs, the model is enforced to embed the perturbed graphs apart from the original graph (Figure 3 (b)). Therefore, the model learned by our discrimination scheme will be capable of distinguishing even the slight differences between the original graph and its perturbations, as well as be capable of capturing the correct distribution of the graphs, since perturbed graphs could be semantically incorrect.

**Perturbation** The remaining question is how to perturb the graph to generate negative examples (i.e., $\mathcal{G}_i$). We aim at adding or deleting the *subset* of edges in the given graph, and thus such perturbed graphs are slightly different from the original graph but also could be semantically incorrect. Again, in this way, since our graph discrimination objective in Equation 4 discriminates the original graph from slightly perturbed graphs, the graph-level representations from our pre-trained GNNs can capture subtle difference that may have a large impact on the properties of graphs, for downstream tasks.

In particular, our perturbation scheme consists of the following two steps: 1) removing and adding a small number of edges, and 2) masking node attributes (Figure 2). To be specific, given an original graph $\mathcal{G} = (\mathcal{V}, \mathcal{E}, \mathcal{X}_{\mathcal{V}}, \mathcal{X}_{\mathcal{E}})$, we aim to perturb it $n$ times, to obtain $\{\mathcal{G}_1, ..., \mathcal{G}_n\}$. To do so, we first manipulate the edge set $\mathcal{E}$ by removing existing edges as well as adding new edges on it and then adjust its corresponding edge matrix $\mathcal{X}_{\mathcal{E}}$. After that, we further randomly mask the node attributes on $\mathcal{X}_{\mathcal{V}}$ for both original and perturbed graphs, to make it more difficult to distinguish between them. Formally, the original and perturbed graphs from our node/edge perturbations are obtained as follows:

$$\mathcal{G}_0 = \left(\mathcal{V}, \mathcal{E}, \tilde{\mathcal{X}}_{\mathcal{V}}^0, \mathcal{X}_{\mathcal{E}}\right), \ \tilde{\mathcal{X}}_{\mathcal{V}}^0 \sim \mathtt{M}(\mathcal{G}), \tag{5}$$

$$\mathcal{G}_i = \left(\mathcal{V}, \mathcal{E}^i, \tilde{\mathcal{X}}_{\mathcal{V}}^i, \mathcal{X}_{\mathcal{E}}^i\right), \ \tilde{\mathcal{X}}_{\mathcal{V}}^i \sim \mathtt{M}(\mathcal{G}), \ (\mathcal{E}^i, \mathcal{X}_{\mathcal{E}}^i) \sim \mathtt{P}(\mathcal{G}),$$

where $\mathtt{M}$ and $\mathtt{P}$ are the node masking and edge perturbation functions, respectively. The scores of original and perturbed graphs are then computed with the score function $f_S$: $[S_0, S_1, ..., S_n] = [f_S(\mathcal{G}_0), f_S(\mathcal{G}_1), ..., f_S(\mathcal{G}_n)]$, which are then used for our previous learning objective in Equation 4.

## 3.3 Learning Discrepancy with Edit Distance

While the learning to discriminate original and perturbed graphs described in Section 3.2 allows the model to learn the discrepancy between original and perturbed graphs by embedding them onto different points, it cannot learn *how dissimilar* the perturbed graph is from the original one as all perturbed graphs are considered as equal (Figure 3 (b)). Thus, to further learn the accurate discrepancy among different graphs, here we introduce a method for efficiently computing the graph discrepancy between original and perturbed graphs leveraging the graph edit distance [23]. Then we describe how to preserve the calculated discrepancy in the learned graph representation space.

**Graph Edit Distance** To preserve the exact amount of discrepancies between original and perturbed graphs in the learned representation space, we first need to measure the graph distance. Graph edit distance, which is widely used to measure the dissimilarity between two graphs [23], is defined by the number of insertion, deletion, and substitution operations for nodes and edges, to transform one graph into the other graph. Although computing

Table 1: Costs for calculating distance, with $m$ edges and $h$ iterations for WL-kernel [25].

| Methods | Costs |
|---|---|
| Graph edit distance [23] | NP-hard [44] |
| Weisfeiler-Lehman kernel [25] | $\mathcal{O}(hm)$ |
| Graph edit distance **with our perturbation** | $\mathcal{O}(1)$ |

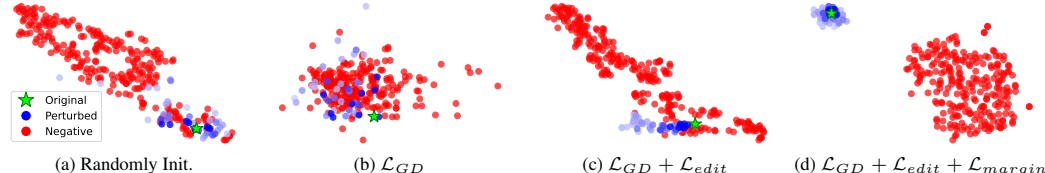

| (a) Randomly Init. | (b) $\mathcal{L}_{GD}$ | (c) $\mathcal{L}_{GD} + \mathcal{L}_{edit}$ | (d) $\mathcal{L}_{GD} + \mathcal{L}_{edit} + \mathcal{L}_{margin}$ |

Figure 3: Visualizations of the graph embeddings to see the effect of each loss for our D-SLA. We represent the level of perturbations via the transparency of color, i.e., the stronger the perturbation, the lighter the color.

the graph edit distance is the NP-hard problem [44], for our case, it is trivially obtained without any cost. This is because, for each perturbed graph, we know the exact number of edge addition and deletion steps as we have generated it using the same process. In other words, the number of added and deleted edges is simply the edit distance between original and perturbed graphs. This brings us significant computational advantages, as we generally require a significant amount of costs for calculating or estimating the distances between two graphs, as shown in Table 1. Note that although several existing works [18, 3] leverage graph edit distance, they are completely different from our work since they are neither self-supervised learning methods nor leverage the graph perturbation.

**Distance-based Discrepancy Learning** Based upon the graph edit distance, we design the regularization term to learn the exact amount of differences between original and perturbed graphs over the embedding space. To be specific, we regularize the model to learn that the embedding-level difference between original and perturbed graphs is proportional to their actual graph edit distance (i.e., if the edit distance between two graphs is large, then they are far away in the embedding space, whereas, if it is small, then they are close to each other) as shown in Distance-based Learning of Figure 2.

We first let an original graph $\mathcal{G}_0$ be an anchor graph, then a graph edit distance and an embedding-level distance between the anchor graph $\mathcal{G}_0$ and the perturbed graph $\mathcal{G}_i$ are defined as $e_i$ and $d_i$, respectively. With such notations, to achieve our objective of learning an exact amount of discrepancy across original and perturbed graphs, we formalize our edit distance-based regularization loss as follows:

$$\mathcal{L}_{edit} = \sum_{i,j} \left( \frac{d_i}{e_i} - \frac{d_j}{e_j} \right)^2 \text{ with } d_i = f_{\text{diff}}(h_{\mathcal{G}_0}, h_{\mathcal{G}_i}), \tag{6}$$

where $f_{\text{diff}}$ measures the embedding-level differences between graphs with $L_2$-norm. We find that Equation 6 can capture the exact amount of differences as shown in Figure 3 (c). Note that Equation 6 alone does not work as shown in Table 5, since the trivial solution of Equation 6 is to set all the embedding-level distance between original and perturbed graphs as zero (i.e., $d_i = 0 \; \forall i$). However, learning the accurate amount of discrepancy without the trivial solution is feasible by jointly training with the objective of original graph discrimination in Equation 4 where the model should differently embed $\mathcal{G}_0$ and $\mathcal{G}_i$ to discriminate them. Furthermore, as we can generate different levels of perturbations according to the graph edit distance, our graph embedding space can interpolate between two different graphs, for example, weakly and strongly perturbed graphs in terms of their actual distances, unlike previous predictive and contrastive learning methods.

### 3.4 Relative Discrepancy Learning with Other Graphs

Thus far, we aimed toward distinguishing the perturbed graphs from the original graph in Section 3.2, with their actual edit distance-based discrepancy learning in Section 3.3. However, such schemes alone will not allow the model to learn the discrepancy between two completely different graphs (Figure 3 (c)). While one might consider learning the discrepancy between original and its negative graphs (other graphs in the same batch) with their graph edit distance, it is impractical as computing the edit distance between completely different graphs is NP-hard [44]. Thus, we instead propose to learn a relative distance, exploiting the assumption that the distance between original and negative graphs in the same batch is larger than the distance between original and perturbed graphs with some amount of *margin* (See Figure 2, Distance-based Learning). The usage of margin is highly beneficial for our discrepancy-based learning framework, since, if the negative graphs are far apart than the amount of margin plus the distance between original and perturbed graphs, the model does not attract the perturbed graphs to the original graph, therefore not losing the discrepancy learned in Equation 6.

Formally, we realize the relative discrepancy learning with the triplet margin loss as follows:

$$\mathcal{L}_{margin} = \sum_{i,j} \max(0, \; \alpha + d_i - d'_j), \tag{7}$$

where $d_i$ is the distance between original and its perturbed graphs: $d_i = f_{\text{diff}}(h_{\mathcal{G}_0}, h_{\mathcal{G}_i})$, $d'_j$ is the distance between original and its negative graphs: $d'_j = f_{\text{diff}}(h_{\mathcal{G}_0}, h_{\mathcal{G}'_j})$ with $\mathcal{G}'$ as in-batch negative graphs, and $\alpha > 0$ as a margin hyperparameter. In this way, we can allow the model to learn that, for the original graph, negative graphs are more dissimilar than the perturbed graphs, while perturbed graphs are also marginally dissimilar to it with the amount of edit distances as shown in Figure 3 (d).

## 3.5 Overall Framework

To sum up, our D-SLA framework aims to learn a graph representation space which preserves the discrepancy among graphs. Our graph discriminator described in Section 3.2 helps learn a space that can discriminate even the slightly perturbed graph from its original graph (Figure 3 (b)). Then, the discrepancy learning with graph edit distance described in Section 3.3 (Figure 3 (c)) enforces the embedding space to preserve the exact discrepancy among graphs. Finally, the margin-based triplet constraints between the perturbed and completely different graphs in Section 3.4 allow the learned embedding space to capture the relative distance between graphs (Figure 3 (d)). Our overall learning objective, dubbed as Discrepancy-based Self-supervised LeArning (D-SLA), is given as follows:

$$\mathcal{L} = \mathcal{L}_{GD} + \lambda_1 \mathcal{L}_{edit} + \lambda_2 \mathcal{L}_{margin}, \tag{8}$$

where hyperparameters $\lambda_1$ and $\lambda_2$ are scaling weights to each loss. Note that, unlike predictive learning [11] which aims to capture local semantics of graphs by masking and predicting local components, such as nodes and edges, our D-SLA learns the graph-level representations where the subtle differences on corner regions of different graphs could be captured by distinguishing original and perturbed graphs. Moreover, our D-SLA can discriminate differently perturbed graphs having different properties, unlike contrastive learning [42, 43] that considers them as similar.

# 4 Experiments

In this section, we first experimentally validate the proposed Discrepancy-based Self-supervised LeArning (D-SLA) on graph classification tasks to verify its effectiveness in obtaining accurate graph-level representations. After that, we further evaluate our D-SLA on link prediction tasks for which capturing the local semantics of graphs is important.

## 4.1 Graph Classification

Accurately capturing the global semantics of given graphs is crucial for graph classification tasks, on which we validate the performance of our D-SLA against existing baselines.

**Experimental Setup** We evaluate our D-SLA on two different domains: the molecular property prediction task from chemical domain [34, 10] and the protein function prediction task from biological domain [11]. For the chemical domain, we follow the experimental setup from You et al. [43], whose goal is to predict the molecules' biochemical activities. For the self-supervised learning, we use 2M molecules from the ZINC15 dataset [26]. Then, after the self-supervised learning, we perform fine-tuning on datasets from MoleculeNet [34] to evaluate the down-stream performances of models. For the biological domain, we follow the setup from You et al. [43], for which the goal is to predict the proteins' biological functions. For pre-training and fine-tuning, we use the dataset of PPI networks [47]. We use the ROC-AUC value as an evaluation metric, and report the average performance over five different runs following Xu et al. [39]. See Appendix A.2 for dataset details.

**Models** We compare our D-SLA against predictive learning baselines: EdgePred [8], AttrMasking [11] and ContextPred [11], and contrastive learning baselines: Infomax [32], GraphCL [42], JOAO [43], GraphLoG [39], and BGRL [30]. Detailed explanations of models including baselines and ours are provided in Appendix A.1.

**Implementation Details** We follow the conventional experimental setup of graph self-supervised learning from Hu et al. [11], where we use the GIN [38] as the base network. For pre-training of our model, we perturb the original graph three times. To be specific, to obtain the perturbed graphs from the original graph, we first randomly select a subgraph of it, and then add or remove edges in the range of $\{20\%, 40\%, 60\%\}$ over the sampled subgraph. For fine-tuning, we follow the hyperparameters from You et al. [43]. We provide additional details in Appendix A.2.

Table 2: Fine-tuning results on graph classification tasks of chemical and biological domains. Best performances are highlighted in bold. The reported results are taken from You et al. [43] and Xu et al. [39], except for the BGRL results and the PPI performance of GraphLoG as they are not available.

| | SSL methods | BBBP | ClinTox | MUV | HIV | BACE | SIDER | Tox21 | ToxCast | PPI | Avg. |
|---|---|---|---|---|---|---|---|---|---|---|---|
| | No Pretrain | 65.8 ± 4.5 | 58.0 ± 4.4 | 71.8 ± 2.5 | 75.3 ± 1.9 | 70.1 ± 5.4 | 57.3 ± 1.6 | 74.0 ± 0.8 | 63.4 ± 0.6 | 64.8 ± 1.0 | 66.72 |
| *Predictive* | Edgepred | 67.3 ± 2.4 | 64.1 ± 3.7 | 74.1 ± 2.1 | 76.3 ± 1.0 | 79.9 ± 0.9 | 60.4 ± 0.7 | 76.0 ± 0.6 | 64.1 ± 0.6 | 65.7 ± 1.3 | 69.77 |
| | AttrMaskig | 64.3 ± 2.8 | 71.8 ± 4.1 | 74.7 ± 1.4 | 77.2 ± 1.1 | 79.3 ± 1.6 | 61.0 ± 0.7 | 76.7 ± 0.4 | 64.2 ± 0.5 | 65.2 ± 1.6 | 70.49 |
| | ContextPred | 68.0 ± 2.0 | 65.9 ± 3.8 | 75.8 ± 1.7 | 77.3 ± 1.0 | 79.6 ± 1.2 | 60.9 ± 0.6 | 75.7 ± 0.7 | 63.9 ± 0.6 | 64.4 ± 1.3 | 70.17 |
| *Contrastive* | Infomax | 68.8 ± 0.8 | 69.9 ± 3.0 | 75.3 ± 2.5 | 76.0 ± 0.7 | 75.9 ± 1.6 | 58.4 ± 0.8 | 75.3 ± 0.5 | 62.7 ± 0.4 | 64.1 ± 1.5 | 69.60 |
| | GraphCL | 69.68 ± 0.67 | 75.99 ± 2.65 | 69.80 ± 2.66 | 78.47 ± 1.22 | 75.38 ± 1.44 | 60.53 ± 0.88 | 73.87 ± 0.66 | 62.40 ± 0.57 | 67.88 ± 0.85 | 70.44 |
| | JOAO | 70.22 ± 0.98 | **81.32 ± 2.49** | 71.66 ± 1.43 | 76.73 ± 1.23 | 77.34 ± 0.48 | 59.97 ± 0.79 | 74.98 ± 0.29 | 62.94 ± 0.48 | 64.43 ± 1.38 | 71.07 |
| | JOAOv2 | 71.39 ± 0.92 | 80.97 ± 1.64 | 73.67 ± 1.00 | 77.51 ± 1.17 | 75.49 ± 1.27 | 60.49 ± 0.74 | 74.27 ± 0.62 | 63.16 ± 0.45 | 63.94 ± 1.59 | 71.21 |
| | GraphLoG | 72.5 ± 0.8 | 76.7 ± 3.3 | 76.0 ± 1.1 | 77.8 ± 0.8 | 83.5 ± 1.2 | **61.2 ± 1.1** | 75.7 ± 0.5 | 63.5 ± 0.7 | 66.92 ± 1.58 | 72.65 |
| | BGRL | 66.73 ± 1.70 | 64.74 ± 6.46 | 69.36 ± 2.66 | 75.52 ± 1.85 | 71.27 ± 5.48 | 60.41 ± 1.43 | 74.83 ± 0.74 | 63.20 ± 0.76 | 64.69 ± 1.66 | 67.86 |
| | D-SLA (Ours) | **72.60 ± 0.79** | 80.17 ± 1.50 | **76.64 ± 0.91** | **78.59 ± 0.44** | **83.81 ± 1.01** | 60.22 ± 1.13 | **76.81 ± 0.52** | **64.24 ± 0.50** | **71.56 ± 0.46** | **73.85** |

| Dataset | Act. Sim. | Inact. Sim. |
|---|---|---|
| BACE | **0.6743** | **0.5403** |
| HIV | 0.4186 | 0.4536 |
| MUV | 0.1946 | 0.4181 |
| Tox21 | 0.3047 | 0.3462 |
| ToxCast | 0.2193 | 0.2962 |
| SIDER | 0.2880 | 0.2316 |
| ClinTox | 0.2725 | 0.2278 |
| BBBP | 0.3961 | 0.2031 |

Table 3: Tanimoto similarity on chemical benchmark datasets.

Figure 4: Visualization of molecules in BACE, where the active and inactive molecules in the top and bottom sides are the most similar ones to the molecule in the left, according to the Tanimoto similarity.

**Results** Table 2 shows that our D-SLA achieves the best average performance against existing predictive and contrastive learning baselines on tasks from both chemical and biological domains, demonstrating the effectiveness of our discrepancy-based framework. To better see what aspects of D-SLA contribute to the performance improvements, we perform in-depth analyses of each dataset, and how our model can effectively handle the task on it, in the next paragraph.

**Analysis** To analyze whether the structural similarity of molecules is correlated to their biochemical activities, we measure the inherent discrepancy of graphs with the Tanimoto similarity over Morgan fingerprints [21]. To be specific, we first iteratively sample an anchor molecule among active molecules in the dataset, and then measure the average Tanimoto similarities of the five most similar active/inactive molecules. In other words, the high similarity values of Inact. Sim. in Table 3 suggests that the molecules have highly overlapped structures regardless of their biochemical activities. For example, as shown in Table 3, the molecules in the BACE dataset are highly correlated, although their activities are different. Also, we further observe that, as shown in Figure 4 that visualizes the most similar active/inactive molecules with respect to the certain anchor molecule in the BACE dataset, the structures between active and inactive molecules are highly similar.

From the above observations, we suggest that, due to the discrete nature of graphs, two slightly different graphs can have completely different properties, which may be the reason for the performance degeneration of contrastive learning methods in Table 2 – which consider perturbed graphs as similar while they might not be the semantically same – on such particular datasets (e.g., BACE, MUV, Tox21, ToxCast) having the high Inact. Sim. scores in Table 3. However, our D-SLA largely outperforms contrastive learning baselines on them[2], because ours not only discriminates an original graph from its perturbations but also can learn their exact discrepancy via the graph edit distance. We note that our method shows the competitive performance on ClinTox and SIDER in Table 2, since they have the lowest structural similarities across different biochemical properties, for which contrastive learning could be effective. We provide more analysis in Appendix B.1.

### 4.2 Link Prediction

Accurately capturing the local semantics of a graph is an important requisite for solving node/edge-level tasks. Thus, we further validate our D-SLA on the link prediction task.

**Experimental Setup** We conduct link prediction experiments on social network datasets – COLLAB, IMDB-B, and IMDB-M – from the TU benchmarks [19]. We separate the dataset into four parts: pre-training, training, validation, and test sets in the ratio of 5:1:1:3. We use the average

---

[2]GraphLoG also adopts the predictive learning scheme by matching the locally masked nodes/subgraphs to their original substructures, thus it shows decent performances on such high similarity datasets.

| SSL Method | COLLAB | IMDB-B | IMDB-M | Avg. |
|---|---|---|---|---|
| No Pretrain | 80.01 ± 1.14 | 68.72 ± 2.58 | 64.93 ± 1.92 | 71.22 |
| *Pred* AttrMasking | 81.43 ± 0.80 | 70.62 ± 3.68 | 63.37 ± 2.15 | 71.81 |
| ContextPred | 83.96 ± 0.75 | 70.47 ± 2.24 | 66.09 ± 2.74 | 73.51 |
| *Contra* Infomax | 80.83 ± 0.62 | 67.25 ± 1.87 | 64.98 ± 2.47 | 71.02 |
| GraphCL | 76.04 ± 1.04 | 63.71 ± 2.98 | 62.40 ± 3.04 | 67.38 |
| JOAO | 76.57 ± 1.54 | 65.37 ± 3.23 | 62.76 ± 1.52 | 68.23 |
| GraphLoG | 82.95 ± 0.98 | 69.71 ± 3.18 | 64.88 ± 1.87 | 72.51 |
| BGRL | 76.79 ± 1.13 | 67.97 ± 4.14 | 63.71 ± 2.09 | 69.49 |
| D-SLA (Ours) | **86.21 ± 0.38** | **78.54 ± 2.79** | **69.45 ± 2.29** | **78.07** |

Table 4: Link prediction results on the social network datasets. The reported results are the average precision, and the bold number denotes the best performance on each dataset.

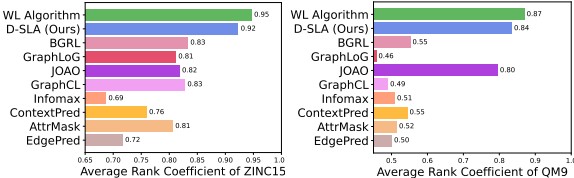

Figure 5: Rank correlation coefficient of 1,000 graphs from ZINC15 and QM9 datasets, measuring the coefficient between the actual similarity ranks and the calculated similarity ranks, between original and perturbed graphs.

precision as an evaluation metric, and report the results over five different runs. We provide more details in Appendix A.3.

**Implementation Details** For GNNs, we use the GCN [15] following You et al. [43]. For perturbation, we add or delete only a tiny amount of edges (e.g., 1 or 2 edges) while increasing the magnitude of perturbation to obtain three perturbed graphs. Note that we do not use the margin triplet loss in Section 3.4, since for local prediction tasks, the graph-level discrepancy learning between completely different graphs is not much helpful in capturing local semantics. For fine-tuning, we train the GNNs to predict whether there is an edge between nodes, and evaluate the GNNs for predicting the existence of 10 edges. For more implementation details, please see Appendix A.3.

**Results** As shown in Table 4, predictive learning baselines outperform other baselines, since it learns to predict local node/edge attributes. However, our D-SLA largely outperforms all baselines, while it aims to discriminate graph-level representations though. We suggest that this is because ours can capture subtle differences of graphs by leveraging the graph edit distance, demonstrating that accurate discrepancy learning is obviously useful for the local link prediction task.

## 4.3 Analysis

In this section, we further analyze the efficacy of our D-SLA. We provide the additional experimental details in Appendix A.4.

**Rank Correlation Coefficient** To see whether learned representations capture the exact amount of discrepancy, we compare the ranks of the calculated vs actual similarities between original and perturbed graphs with Spearman's rank correlation coefficient. Note that the results of the WL algorithm [17] are merely a performance indicator of discriminative power since it cannot obtain representations that generalize to downstream tasks. As shown in Figure 5, unlike baselines that mostly fail to discriminate different graphs, our D-SLA has the discriminative power that is on par with the powerful WL algorithm, while ours can generalize to downstream tasks as shown in Table 2.

**Ablation Study** To see how much each component contributes to the performance gain, we conduct an ablation study. As shown in Table 5, we observe that our graph discrimination task ($\mathcal{L}_{GD}$) significantly improves the performance on down-stream tasks against no-pretraining. However, leveraging the exact edit distance alone ($\mathcal{L}_{edit}$) does not learn meaningful representations, since the model trivially sets all the distances between original and its perturbations as zero, as discussed in Section 3.3. Also, we further observe that

Table 5: Ablation study for our D-SLA on graph classification and link prediction tasks.

| $\mathcal{L}_{GD}$ | $\mathcal{L}_{edit}$ | $\mathcal{L}_{margin}$ | ClinTox | BACE | COLLAB |
|---|---|---|---|---|---|
| | | | 58.00 | 70.10 | 71.21 |
| ✓ | | | 70.83 | 81.58 | 74.23 |
| | ✓ | | 57.46 | 69.99 | 72.61 |
| ✓ | ✓ | | 77.48 | 83.53 | 76.19 |
| ✓ | ✓ | ✓ | 80.17 | 83.81 | N/A |

each component of distance-based learning in Section 3.3 and 3.4 helps to improve the performance, verifying that accurate discrepancy learning with edit and relative distances is important for modeling graphs. We further discuss the ablation study in Appendix B.2.

**Embedding Visualization** We visualize the embedding space from various pre-training methods: predictive, contrastive, and ours, in Figure 1 (a,b,c-2). We observe that predictive learning cannot capture the global graph-level similarity well, as it aims to predict subgraphical semantics of graphs during the pretext task. While contrastive learning can closely embed the highly similar graphs to the original graph, it cannot accurately capture the exact amount of discrepancies among perturbed graphs. Contrarily, our D-SLA can accurately distinguish between the original, perturbed, and negative graphs, while accurately capturing the exact amount of discrepancy for the perturbed graphs.

## 5   Conclusion

In this work, we focused on the limitations of existing self-supervised learning for GNNs: predictive learning does not capture the graph-level similarities; contrastive learning might treat two semantically different graphs from perturbations as similar. To overcome such limitations, we proposed a novel framework (D-SLA) that can learn the graph-level differences among different graphs while also can learn the slight edge-wise differences, by discriminating the original from perturbed graphs. Further, the model is trained to differentiate the target graph from its perturbations and other graphs, while preserving the accurate graph edit distance, allowing the model to discriminate between not only two structurally different graphs but also similar graphs with slight differences. We validated our D-SLA on 12 benchmark datasets, achieving the best average performance. Further analysis shows that it learns a discriminative space of graphs, reflecting the graph edit distances between them.

## 6   Limitation and Potential Societal Impacts

In this section, we discuss the limitation and potential societal impacts of our work.

**Limitation**   In this work, we propose a graph self-supervised learning framework, which aims to learn the discrepancy between the original and perturbed graphs, by discriminating the original graph among the original and its perturbations, but also by learning the exact amount of discrepancy across them with the graph edit distance. Note that, while our perturbation scheme can easily make the slight structural differences, it cannot be aware of the semantic differences across differently perturbed graphs, since the graph semantics depend on the target domain. In other words, nodes and edges could differently contribute to the graph semantics depending on the target graph domain (e.g., molecular graphs or social networks), which are hard to pre-define in one framework. As manually annotating the semantic differences across different graphs is impossible to do in an end-to-end fashion, one may need to devise a clever way to be aware of the semantics of graphs for further reflecting the exact semantic difference in the embedding space, which we leave as future work.

**Potential Societal Impacts**   Discovering de novo drugs is significantly important to our society since they can be used for curing a disease or enhancing agricultural production, that is directly related to our life. However, it is extremely costly and time-consuming to design drug candidates and validate them, requiring immense lab experiments and labor. In this work, we verify our proposed method in the graph classification task in Table 2 of the main paper, and ours outperforms the other baselines for classifying toxicity on the Tox21 and ToxCast datasets. We strongly believe that such aspects of our method can positively contribute to our society in developing valuable drugs. However, when validating the toxicity of the most acceptable drug candidates, someone might badly use our method to reduce the money and time (i.e., does not check the toxicity of drugs for reducing costs, however, which may be used for clinical trials to humans), which can negatively affect our society. We sincerely hope that our method would not be utilized for such a bad purpose.

## 7   Acknowledgements and Disclosure of Funding

We thank anonymous reviewers for their constructive feedback. This work was supported by Institute of Information & communications Technology Planning & Evaluation (IITP) grant funded by the Korea government (MSIT) (No.2019-0-00075, Artificial Intelligence Graduate School Program (KAIST), and No.2021-0-02068, Artificial Intelligence Innovation Hub), the Engineering Research Center Program through the National Research Foundation of Korea (NRF) funded by the Korean Government MSIT (NRF-2018R1A5A1059921), Samsung Electronics (IO201214-08145-01), and the HPC Support Project funded by the Ministry of Science and ICT and NIPA.

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
