# [Appendix]
# Graph Self-supervised Learning
# with Accurate Discrepancy Learning

**Organization**   In Section A, we first introduce the baselines and our model and then describe the experimental details of graph classification and link prediction tasks but also our in-depth analyses. Then, in Section B, we provide the additional experimental results about analyses on datasets, ablation study for our proposed objectives, effects of our hyperparameters ($\lambda_1$, $\alpha$, $\lambda_2$, and the perturbation magnitude), ablation study of attribute masking, and the comparison with augmentation-free approaches.

## A   Experimental Details

In this section, we first introduce the computing resources that we use, the baselines, and our model in Section A.1. After that, we describe the experimental setups of the graph classification and link prediction tasks in Section A.2 and Section A.3 as well as the analysis in Section A.4.

**Computing Resources**   For all experiments, we use PyTorch and PyTorch Geometric libraries [7, 1], for easy usage of GPU resources. We use TITAN XP and GeForce RTX 2080 Ti for training and evaluating all models.

### A.1   Baselines and Our Model

1. **EdgePred** is a predictive learning baseline adopted from the link prediction task of Hamilton et al. [2], whose goal is to predict the existence of edges between the given two nodes.
2. **AttrMasking** [3] is a predictive learning baseline that predicts the attributes of masked nodes and edges from the embeddings of nodes.
3. **ContxtPred** [3] is a predictive learning baseline that first samples two different subgraphs from the same centered node, and then trains them to be similar while the subgraphs from the other graphs are trained to be dissimilar.
4. **Infomax** [11] is a contrastive learning baseline, whose goal is to learn the representations for the given graph and the substructure within the same given graph to be similar while learning the representations for the given graph and the substructures from the negative graphs to be dissimilar.
5. **GraphCL** [16] is a contrastive learning baseline, whose goal is to learn the similarity between two perturbed graphs from the same graph contrasting to in-batch negative graphs over the global graph-level representations. In particular, this method uses the following four perturbation methods: attribute masking, edge perturbing, node dropping, and subgraph sampling.
6. **JOAO** [17] is a contrastive learning baseline that, while the learning objective of it is the same as the GraphCL model described above, learns to automatically select the perturbation schemes.
7. **JOAOv2** [17] is a variant of JOAO, which has individual projection heads according to the perturbation schemes. Specifically, a perturbed graph is fed into the typical projection head according to the selected perturbation.
8. **GraphLoG** [15] is a baseline that has two learning objectives: 1) it matches the masked nodes/graphs to their unmasked counterparts; 2) it clusters a group of globally similar graphs with learnable cluster prototypes.
9. **BGRL** [9] is a baseline that maximizes the similarity between two perturbed graphs from the original graph without considering in-batch negative graphs, aiming to represent large-scale graphs with efficiency in memory usage.
10. **D-SLA** is our discrepancy-based graph self-supervised learning framework, which aims to learn the accurate discrepancy between original, perturbed, and negative graphs, by not only discriminating the original graph from its perturbations but also preserving the accurate amount of discrepancy with the graph edit distance between them.

## A.2 Graph Classification

**Datasets** We use the available benchmark datasets[1] for the graph classification task. Specifically, for the chemical domain, we use 2M molecules sampled from the ZINC15 dataset [8] without using any labels on it. The fine-tuning datasets consist of the molecular graphs from MoleculeNet [12], where the classes are given by the biophysical and physiological properties of the molecules. For the biological domain, the datasets are constructed by the sampled ego-networks from the PPI networks [19]. In particular, the pre-training dataset consists of 306K unlabeled protein ego-networks of 50 species, and the fine-tuning dataset consists of 88K protein ego-networks of 8 species with the label given by the functionality of the ego protein. We report the statistics of graph classification datasets in Table S1.

Table S1: Dataset statistics on chemical and biological domains.

| Dataset | Tasks | Graphs | Avg. Nodes | Avg. Edges |
|---|---|---|---|---|
| *Chemical Domain* | | | | |
| ZINC15 (Pre-training) | - | 2,000,000 | 26.62 | 28.86 |
| QM9 (Rank Coeff.) | - | 133,149 | 8.80 | 9.40 |
| BBBP | 1 | 2,039 | 24.06 | 25.95 |
| ClinTox | 2 | 1,478 | 26.16 | 27.88 |
| MUV | 17 | 93,087 | 24.23 | 26.28 |
| HIV | 1 | 41,127 | 25.51 | 27.47 |
| BACE | 1 | 1,513 | 34.09 | 36.86 |
| SIDER | 27 | 1,427 | 33.64 | 35.36 |
| Tox21 | 12 | 7,831 | 18.57 | 19.29 |
| ToxCast | 617 | 8,575 | 18.78 | 19.26 |
| *Biological Domain* | | | | |
| PPI (Pre-training) | - | 306,925 | 39.83 | 364.82 |
| PPI (Fine-tune) | 40 | 88,000 | 49.35 | 445.39 |

**Strategy for Selecting Edges for Perturbations** In this paragraph, we describe the detailed edge selection scheme for our graph perturbation. In our experiments of graph classification, we first select the node and then sample the 3-hop subgraph of it. After that, we randomly add and remove edges on the subgraph. The reason behind selecting the target subgraph for perturbation is that we aim to reduce the potential risk of making unreasonable cycles, which are impractical especially on the chemical domain. Therefore, to prevent the model to learn such an incorrect bias in the embedding space, we rather sample the subgraph for perturbing the edges.

**Common Implementation Details** We follow the conventional design choice of GNNs for evaluating the graph self-supervised learning methods from Hu et al. [3]: Graph Isomorphism Networks (GINs) [14] consisting of 5 layers with 300 dimensions along with mean average pooling for obtaining the entire graph representations. For pre-training of our D-SLA, we sample a subgraph by randomly selecting a center node and then select 3-hop neighbors of it, and then remove the edges on the selected subgraph three times with different magnitudes (20%, 40%, 60%) to make three perturbed graphs, while memorizing the number of deleted and added edges to calculate the graph edit distance. To prevent the situation where the deleted edges are added again, we add edges that are not present in the given original graphs. We mask 80% of nodes in the selected subgraph to confuse the model to distinguish the original graph from its perturbed graphs. Furthermore, we include the strong perturbation, where 80% of edges are perturbed and 80% of nodes are masked among entire nodes and edges in the given graph. $\lambda_1$ and $\lambda_2$ are set as 0.7 and 0.5, respectively.

**Implementation Details on Molecular Property Prediction** We follow the conventional molecule representation setting from Hu et al. [3], where the node attributes contain the atom number along with the chirality, and the edge attributes contain the bond type (e.g., Single, Double, Triple or Aromatic) along with the bond direction which is represented if an edge is a double or aromatic bond. When adding an edge during edge perturbation, we sample its type by following the distribution of edge attributes in the pre-training dataset. Specifically, we first sample the bond type following the distribution and then sample also the bond direction depending on the bone type. For pre-training, we use the batch size of 256, the number of epochs of 100, the learning rate in the range of [0.01, 0.001, 0.0001], and the margin $\alpha$ in the range of [3,4,5,6,7] by grid search. For the splitting of fine-tuning datasets, we use the scaffold splitting following the conventional setting from Hu et al. [3] and You et al. [17]. For fine-tuning, we also follow the conventional setting from You et al. [17].

**Implementation Details on Protein Function Prediction** We use the pre-defined biological graphs from Hu et al. [3], where a node corresponds to a protein without any attributes, and an edge corresponds to a relation type between two proteins such as biological interaction or co-expression. As in molecular property prediction, we add reasonable edges by following the distribution of edge attributes in the pre-training dataset. For pre-training, the number of epochs is 100, the batch size is 128, the learning rate is 0.001, and the margin is 10. For data splitting of the fine-tuning dataset, we use the provided conventional setting from Hu et al. [3]. For fine-tuning, we also follow the conventional setting from Hu et al. [3]. Note that, as the result of GraphLoG [15] on this protein function prediction task is not available in the referred paper, we produce the result by following the experimental setups along with the provided public source code.

---

[1]http://snap.stanford.edu/gnn-pretrain/data/

## A.3 Link Prediction

**Datasets** The datasets[2] we used for the link prediction task are COLLAB, IMDB-B (IMDB-BINARY), IMDB-M (IMDB-MULTI) – the social network datasets from TU dataset benchmark [6]. COLLAB dataset consists of ego-networks extracted from public scientific collaboration networks, namely High Energy Physics, Condensed Matter Physics, and Astro Physics. IMDB-B and IMDB-M are movie collaboration ego-networks where a node represents an actor/actress. The statistics of social network datasets are provided in Table S2.

Table S2: Statistics of social network datasets used in link prediction experiments.

| Dataset | Graphs | Avg. Nodes | Avg. Edges | Pert. Strength |
|---|---|---|---|---|
| COLLAB | 4320 | 76.12 | 2331.37 | 0.1% |
| IMDB-B | 2039 | 20.13 | 85.48 | 1% |
| IMDB-M | 1478 | 16.64 | 77.90 | 1% |

**Strategy for Selecting Edges for Perturbations** To capture the fine-grained local semantics, we suggest that the weaker magnitude of perturbation is the better (See Section B.6 verifying the effect of edge perturbation strengths). Therefore, we only perturb the tiny amount of edges (e.g., 1 or 2 edges), as shown in Table S2, rightmost column.

**Implementation Details** We use the Graph Convolutional Network (GCN) [5] consisting of three layers with 300 hidden dimensions. Following the previous works [16, 17], we let the node attributes correspond to the degree of the node. For pre-training, we remove the complete graphs – that always have the edges between any two nodes – as we cannot include additional edges during perturbation. For node masking used in AttrMaksing and our D-SLA, we replace the node attribute with the masked token. For hyperparameters, we use the learning rate of 0.001, the batch size of 32, and the $\lambda_1$ of 0.7. During pre-training of our D-SLA, we generate three perturbed graphs by increasing the perturbation magnitudes (e.g., 1%, 2%, 3%). Also, we further mask 20% of nodes in perturbation.

## A.4 Analysis

**Rank Correlation Coefficient** The spearman's rank correlation coefficient measures the correlation between two rank series in the range from -1 to 1, where the value is 1 if the two rank series are perfectly and monotonically the same. We build the following two rank series to compare: 1) the labeled similarity rank between the original and perturbed graphs using the graph edit distance, and 2) the predicted similarity rank based on the embedding-level distances between original and perturbed graphs from pre-trained models. Specifically, we perturb edges of the entire graph by gradually increasing the magnitude of edge perturbation (i.e., 5%, 10%, 15%, 25%, 35%, 45%, 60%, 75%, 90%), and then label ranks of the perturbed graphs to the original graph according to the graph edit distance. Then, the original and perturbed graphs are fed into the pre-trained models, and the ranks are measured by the embedding-level distance between the original and perturbed graphs. Therefore, if a pre-trained model can capture the exact amount of discrepancy, the rank correlation coefficient would be 1, by locating the embedding of a similar graph (a weakly perturbed graph) closer to the original graph than the embedding of a dissimilar graph (a strongly perturbed graph). We measure the coefficient with randomly sampled 1, 000 different graphs.

For models to calculate the similarity across different graphs, we use the pre-trained model for the graph classification task in our D-SLA. For EdgePred, AttrMasking, ContextPred, Informax, and GraphCL baselines, we use the publicly available pre-trained models[3]. For JOAO and GraphLoG, we use the public source codes[4], to obtain the pre-trained models. The WL algorithm in Figure 5 of the main paper corresponds to a randomly initialized GIN model. In other words, since the GIN is as powerful as the WL test, we denote it as the WL algorithm. We evaluate the above models on two different datasets: ZINC15 [8] and QM9 [6], where statistics of each dataset is provided in Table S1. Note that the QM9 dataset is not used for pre-training, thus we can measure the model's generalization ability with it.

**Embedding Visualization** To visualize the representation space of the original, perturbed, and negative graphs, we pre-train the models on the subset of the ZINC15 dataset [8] with the perturbation strategy in Section A.2 for obtaining perturbed graphs. Then, after pre-training, we visualize the graph representations by PCA [4] and t-SNE [10] for Figure 1 and Figure 3 of the main paper, respectively.

---

[2]https://chrsmrrs.github.io/datasets/docs/datasets/

[3]https://github.com/snap-stanford/pretrain-gnns, https://github.com/Shen-Lab/GraphCL

[4]https://github.com/Shen-Lab/GraphCL_Automated, https://github.com/DeepGraphLearning/GraphLoG

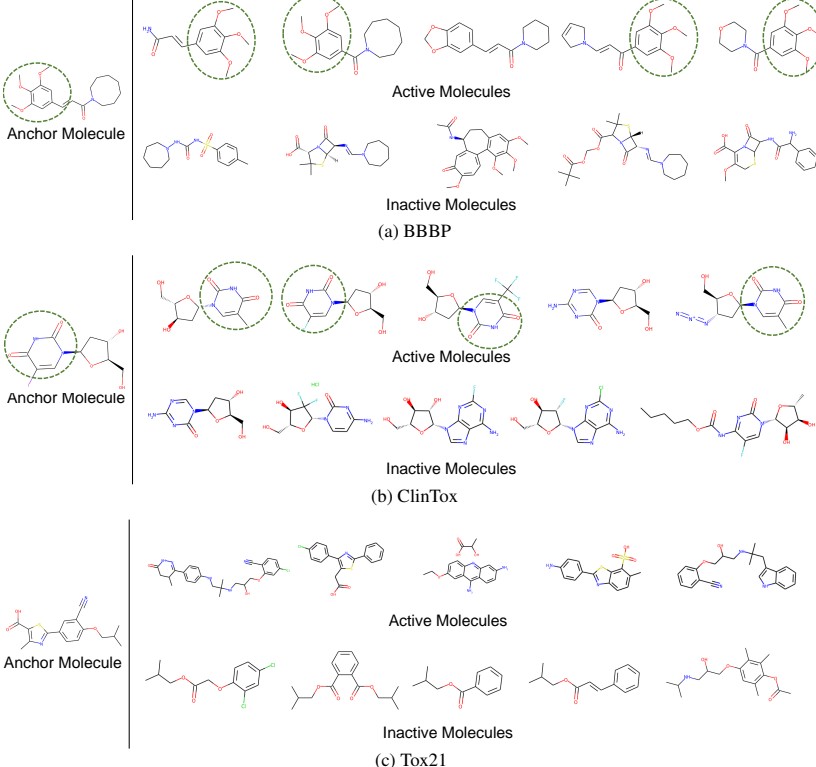

Figure S2: Structural comparisons on the most similar (i.e., top-5) active/inactive molecules for the certain anchor active molecule on the left side, for the BBBP, ClinTox, and Tox21 datasets. Green dotted circles indicate the shared structure across different molecules.

# B    Additional Experimental Results

In this section, we provide additional results with their corresponding discussions. To be specific, in Section B.1, we analyze the correlation between the characteristics of the dataset and self-supervised learning methods. Then, we provide an in-depth discussion about our observations in the ablation study in Section B.2. Additionally, we provide some guidelines for choosing our hyperparameters ($\lambda_1$, $\alpha$, $\lambda_2$, and the perturbation magnitude) in Section B.3, B.4, B.5, and B.6, respectively. Futhermore, we provide an ablation study of attribute masking in Section B.7. Finally, we compare our D-SLA with augmentation-free approaches in Section B.8.

## B.1    Dataset Analysis

In this subsection, we further discuss the characteristics of graph self-supervised learning methods with respect to the characteristics of datasets. As shown in Section 4.1, we find that contrastive learning methods outperform predictive learning methods on BBBP and ClinTox. Contrarily, predictive learning methods outperform contrastive learning methods on Tox21. Therefore, we further analyze BBBP, ClinTox, and Tox21 datasets to answer why such methods have counterfactual effects on different datasets.

In Figure S2, we visualize the structures of active/inactive molecules from the anchor molecule. We observe that in BBBP and ClinTox datasets, the activities are highly correlated to the structural similarity, i.e., the structurally similar molecules show the same activities. Therefore, as contrastive learning aims to maximize the similarity between perturbed graphs from the original graph, it fits into the BBBP and ClinTox datasets, showing better performance than predictive learning methods. However, in the Tox21 dataset, we cannot observe any clues that the activities are correlated to the structural similarity. Therefore, capturing the structural similarity with contrastive learning seems to be useless in this dataset, resulting in the better performance of predictive learning methods. However, our D-SLA can learn the discrete embedding space by learning the discrepancy even between similar graphs, thus obtaining a discriminative space that can further be utilized to distinguish between them for downstream tasks and outperforming all other baselines as shown in Section 4.1.

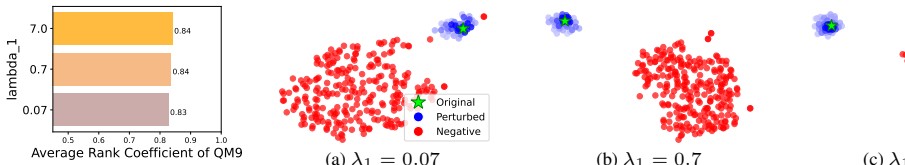

Figure S3: Rank correlation coefficient of QM9 with varying $\lambda_1$ values.

(a) $\lambda_1 = 0.07$      (b) $\lambda_1 = 0.7$      (c) $\lambda_1 = 7.0$

Figure S4: Visualization of the learned latent representation space for different $\lambda_1$ values. Note that each model is trained on the subset of the ZINC15 dataset and the embedding spaces are visualized by t-SNE [10].

## B.2 Additional Interpretation of Ablation Study

We conduct an ablation study on link prediction and graph classification tasks in Table 5 of the main paper. For link prediction, we observe that both two components, $\mathcal{L}_{GD}$ and $\mathcal{L}_{edit}$, consistently improve the performance, thus verifying that our discrepancy-based learning allows the model to capture the local semantics of graphs. Also, for graph classification, we choose the most different two datasets in their properties – ClinTox and BACE datasets – to obviously see the contribution of each component in our D-SLA. In particular, for the ClinTox dataset in which the biochemical activities of molecules are highly correlated to their structures, we observe that it is important to discriminate the negative graphs from the perturbed graphs with the triplet margin loss $\mathcal{L}_{margin}$, as the performance improvements on using it is significant compared to the other dataset: BACE. However, in the case of the BACE dataset, since the molecules are highly similar regardless of their biochemical activities, the graph edit distance loss $\mathcal{L}_{edit}$ largely contributes to the performance gain, allowing the model to learn the exact discrepancy across similar graphs.

## B.3 Effect of $\mathcal{L}_{edit}$ Coefficient ($\lambda_1$)

We demonstrate the effect of the coefficient $\lambda_1$ for $\mathcal{L}_{edit}$ (Equation 8). Specifically, we show its efficacy by measuring the rank correlation coefficient, and by visualizing the graph representation space. Firstly, as shown in Figure S3, the differences in rank correlation with varying scaling coefficient $\lambda_1$ are marginal. Also, as shown in Figure S4, for all models, the perturbed graphs are well embedded along their perturbation magnitudes regardless of $\lambda_1$ values. Therefore, we suggest that, fortunately, learning the accurate amount of discrepancy between graphs does not heavily depend on the scaling hyperparameter for $\mathcal{L}_{edit}$, namely $\lambda_1$. On the other hand, we observe that, if $\lambda_1$ is relatively small, some negative graphs are embedded closer to the perturbed ones (Figure S4 (a)). This result indicates that, the regularization effects of our objective $\mathcal{L}_{edit}$ for learning the discrepancy between the original and perturbed graphs can also affect the boundary between perturbed and negative graphs, as shown in Figure S4 (b), (c).

## B.4 Effect of Margin ($\alpha$) in $\mathcal{L}_{margin}$

To demonstrate the effect of the margin $\alpha$ in $\mathcal{L}_{margin}$ (Equation 7), we pretrain the model with varying $\alpha$ values and fine-tune it on BACE dataset for graph classification. As shown in Table S3, large value of $\alpha$ degenerates our discrepancy learning among similar graphs. As described in Section 3.4, if the distance between the original and its negative graph ($d'_j$) is larger than the distance between the original and perturbed graph ($d_i$) plus $\alpha$ (i.e., $\alpha + d_i < d'_j$), the distance between the original and perturbed graph ($d_i$) is preserved not losing the discrepancy learned by $\mathcal{L}_{edit}$ (Equation 6). However, large $\alpha$ makes it hard to satisfy the condition (i.e., $\alpha + d_i < d'_j$) forcing the model to lose the discrepancy learned by $\mathcal{L}_{edit}$.

Table S3: Effect of $\alpha$ on BACE dataset.

| $\alpha$ | ROC-AUC |
|---|---|
| 1.0 | $83.75 \pm 0.96$ |
| 5.0 | $83.81 \pm 1.01$ |
| 10.0 | $78.34 \pm 1.07$ |

## B.5 Effect of $\mathcal{L}_{margin}$ Coefficient ($\lambda_2$)

We demonstrate the effect of the coefficient $\lambda_2$ for $\mathcal{L}_{margin}$ (Equation 8) by pre-training with various $\lambda_2$ values and fine-tuning on BACE dataset for graph classification. As shown in Table S4, when $\lambda_2$ is large, the performance on BACE downstream dataset is generated, indicating that the model cannot learn the discrepancy among similar graphs. We suggest that this is because $\lambda_2$ controls the intensity of attracting the similar graphs of $\mathcal{L}_{margin}$ and a large value of $\lambda_2$ which would strongly attract the similar graphs forces the model to lose the discrepancy among similar graphs learned by $\mathcal{L}_{edit}$ (Equation 6).

Table S4: Effect of $\lambda_2$ on BACE dataset.

| $\lambda_2$ | ROC-AUC |
|---|---|
| 0.1 | $83.68 \pm 0.78$ |
| 0.5 | $83.81 \pm 1.01$ |
| 0.9 | $80.72 \pm 0.71$ |

Table S7: Fine-tuning results on graph classification tasks. Best performances are highlighted in bold.

| SSL methods | BBBP | ClinTox | MUV | HIV | BACE | SIDER | Tox21 | ToxCast | Avg. |
|---|---|---|---|---|---|---|---|---|---|
| SimGCL [18] | 67.37 ± 1.23 | 55.66 ± 4.72 | 71.24 ± 1.79 | 75.04 ± 0.86 | 74.11 ± 2.74 | 57.44 ± 1.74 | 74.39 ± 0.45 | 62.27 ± 0.38 | 67.19 |
| SimGRACE [13] | 71.25 ± 0.86 | 64.16 ± 4.50 | 71.18 ± 3.40 | 74.52 ± 1.12 | 73.81 ± 1.37 | **60.59** ± 0.96 | 74.20 ± 0.64 | 63.36 ± 0.52 | 69.13 |
| D-SLA (Ours) | **72.60** ± 0.79 | **80.17** ± 1.50 | **76.64** ± 0.91 | **78.59** ± 0.44 | **83.81** ± 1.01 | 60.22 ± 1.13 | **76.81** ± 0.52 | **64.24** ± 0.50 | **74.51** |

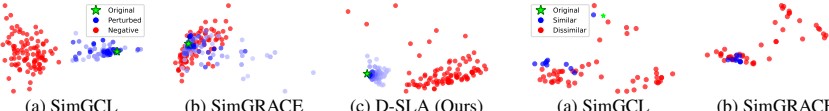

(a) SimGCL    (b) SimGRACE    (c) D-SLA (Ours)    (a) SimGCL    (b) SimGRACE    (c) D-SLA (Ours)

Figure S5: Embedding space visualization on similar and dissimilar graphs with Graph Edit Distance.

Figure S6: Embedding space visualization on similar and dissimilar graphs with Tanimoto Similarity.

## B.6 Effect of Perturbation Magnitude on Link Prediction

We validate the effect of the perturbation magnitude on the COLLAB dataset for link prediction. As shown in Table S5, we observe that the performance of link prediction is enhanced when only a small amount of edges are perturbed, demonstrating that weaker perturbation magnitude is better for capturing local semantics. If the perturbation magnitude is weak, the perturbed graphs are slightly different from the original graph, thus the model could capture a subtle difference across original and perturbed graphs.

Table S5: Effect of magnitude of perturbation on the link prediction task.

| Magnitude | Accuracy |
|---|---|
| 10% | 70.67 ± 0.63 |
| 1% | 74.55 ± 0.76 |
| 0.1% | 76.19 ± 0.50 |

## B.7 Ablation Study of Attribute Masking

We conduct an additional ablation study for the attribute masking in our perturbation strategy on the COLLAB dataset for link prediction. As shown in Table S6, the performance without attribute masking is significantly lower than the performance with attribute masking. We suggest that, in the pre-training stage, attribute masking limits the information given to the model and forces the model to learn more transferable and fruitful representations, demonstrating that attribute masking in our perturbation is a key factor to learn the local semantics.

Table S6: Ablation study of attribute masking on the link prediction task.

| | Accuracy |
|---|---|
| w/ Masking | 76.19 ± 0.50 |
| w/o Masking | 70.42 ± 0.95 |

## B.8 Comparison with Augmentation-Free Contrastive Learning Approaches

Recently, augmentation-free contrastive learning methods have been proposed. We validate the effectiveness of our discrepancy learning framework by comparing with augmentation-free approaches on graph classification and link prediction tasks. We compare our D-SLA with SimGCL [18] and Sim-

Table S8: Fine-tuning results on link prediction tasks. Best performances are highlighted in bold.

| | COLLAB | IMDB-B | IMDB-M | Avg. |
|---|---|---|---|---|
| SimGCL | 77.46 ± 0.86 | 64.91 ± 2.60 | 63.78 ± 2.28 | 68.72 |
| SimGRACE | 74.51 ± 1.54 | 64.49 ± 2.79 | 62.81 ± 2.32 | 67.27 |
| D-SLA (Ours) | **86.21** ± 0.38 | **78.54** ± 2.79 | **69.45** ± 2.29 | **78.07** |

GRACE [13] that augment views of graphs by adding noise to graph embeddings or model parameters while preserving the graph structures. As shown in Table S7 and Table S8, our discrepancy learning outperforms the augmentation-free contrastive learning approaches, demonstrating the effectiveness of our discrepancy learning framework in capturing both local and global semantics. We further visualize the embedding space of similar and dissimilar graphs with different distance metrics such as Graph Edit Distance (Figure S5) and Tanimoto similarity (Figure S6). We observe that, in SimGCL and SimGRACE, the similar and dissimilar graphs are not distinguished and the augmentation-free approaches cannot capture the exact amount of discrepancy, since they cannot learn the difference between similar graphs. Contrarily, by our discrepancy learning, the model can distinguish similar and dissimilar graphs and learn the exact amount of discrepancy both on Graph Edit Distance and Tanimoto Similarity.