# OpenReview forum: "Graph Self-supervised Learning with Accurate Discrepancy Learning"
_NeurIPS.cc/2022/Conference — NeurIPS 2022 Accept_

### Official Review · Reviewer_NxMJ · 2022-06-22

**Rating:** 4
**Confidence:** 4
**Soundness:** 3 good
**Presentation:** 3 good
**Contribution:** 2 fair

**Summary:**

This paper studies the problem of self-supervised learning of graph neural networks. The goal is to learn representations for nodes and graphs in an unsupervised manner. The paper studies the limitations of the predictive and contrastive learning methods and proposes a new framework (D-SLA) to incorporate the discrepancy between the original and perturbed graphs in the self-supervised loss. D-SLA is tested for various tasks.

**Questions:**

1. Tables 2 and 4: how many times did the authors run the model to get the results in these two tables? which results are statistically significant compared to the best baseline?

2. How does the value of $\lambda_1$ and $\lambda_2$ in Equation 8 affect the performance of the model? some charts that show the performance of the model for different values of $\lambda_1$ and $\lambda_2$ will be helpful.

3. The paper tested the proposed model for link prediction on social networks. However, I find the motivation of the paper to be weak in social networks (and stronger for small graphs like molecules). Why some perturbations in a large social network might drastically change some of its properties?

**Limitations:**

See weaknesses 1 and 2.

**Strengths And Weaknesses:**

Strengths:

1. Self-supervised learning is an important problem that has recently gained a lot of attention.

2. The proposed method is tested on various tasks on several graph benchmarks.


Weaknesses:


1. The proposed self-supervised loss in Equation 8 mainly focuses on the global information in a graph as Equations 4, 6, and 7 are all defined on the graph embedding level. This might cause the model not to pay attention to local information in the graph.

2. The motivation of the work is based on a pitfall of contrastive learning methods, which assumes two similar graphs (one original and one perturbed version of the original) are the same. The paper proposed using distance-based self-supervised learning which incorporates the distance of two graphs when pushing their embeddings closer or pulling their embeddings apart in the latent space. However, the notion of distance here is a weighted version of the original contrastive learning version:  two similar graphs will still have small distances and be close in the latent space even if some properties of the perturbed graph drastically change from the original graph. Also, it is not clear why a perturbed graph with a distance of two to the original graph should be further apart compared to a graph with a distance of one to the original graph. How can we make sure that graph edit distance is a good proxy for measuring if two graphs should be close or far in the latent space?

---

> ### Author Response · Authors · 2022-08-01
> **Initial Response (5/5) to Reviewer NxMJ**
>
> **Question 6.** How does the value of $\lambda_1$ and $\lambda_2$ in Equation 8 affect the performance of the model? some charts that show the performance of the model for different values of $\lambda_1$ and $\lambda_2$ will be helpful.
>
> **Answer 6.** Thank you for the suggestion. Please note that we already provided the effect of $\lambda_1$ in Appendix B.4, with Figure S3 and Figure S4, where we observed that our D-SLA is not sensitive across different $\lambda_1$ values.
>
> On the other hand, based on your suggestion, we further provide the effect of $\lambda_2$ by first varying its values during pre-training and then measuring the downstream performances on the BACE dataset. Note that, to measure the effectiveness of $\lambda_2$, we particularly chose the BACE dataset, since two structurally similar graphs are often different in their activities as shown in Table 3, thus learning subtle discrepancies is important for this dataset.
> | $\lambda_2$ | ROC-AUC        |
> |-----|----------------|
> | 0.1 | 83.68 ± 0.78 |
> | 0.5 | 83.81 ± 1.01 |
> | 0.9 | 80.72 ± 0.71 |
>
> Table R3. The sensitivity of $\lambda_2$ on BACE dataset finetuning.
>
> As shown in Table R3, we observe that a larger $\lambda_2$ value degenerates the performance of our D-SLA, since too large $\lambda_2$ value forces the model to too focus on learning the discrepancy between two completely different graphs (i.e., perturbed and negative graphs), rather than learning the subtle differences among original and its slightly perturbed graphs.
>
> We further add the effect of varying $\lambda_2$ in the next revision.
>
> ---
>
> **Question 7.** Tables 2 and 4: how many times did the authors run the model to get the results in these two tables?
>
> **Answer 7.** As described in Line 276 and Lines 319-320, we run our experiments five times.
>
> ---
>
> **Question 8.** Which results are statistically significant compared to the best baseline?
>
> **Answer 8.** Thank you for your helpful suggestion in regard to the statistical analysis, from which the significance of our D-SLA becomes clearer. Specifically, we have conducted the t-test with p-value of 0.05 and observed that, compared to all models, our D-SLA achieves statistically significant results on all datasets in the link prediction task. Also, in the graph classification task, our D-SLA achieves statistically significant results on five datasets among the nine compared against the best baseline (i.e., GraphLog) which is not significant on all datasets.

---

> ### Author Response · Authors · 2022-08-01
> **Initial Response (4/5) to Reviewer NxMJ**
>
> (Continued from Answer 4 in the comment box above)
> * **Optimizing Equation (4)**: We train the model by optimizing Equation (4) to learn the local differences with the original graph ($G_0)$ and a graph perturbed by one edge ($G_1$). Here we denote 'Target Node' as a node where an edge perturbation is applied. As shown in Table R1, learning local differences on the graph-level representations **affects local representations only in the subgraphical region** where the perturbation is applied (target nodes and 1-hop neighbor nodes of the target node) and does not affect the local representations of distant nodes from the target node (Most Distant Nodes Distance).
> | Epoch | Graph Distance | Target Node Distance | 1-hop Nodes Distance  | Most Distant Nodes Distance |
> |-------|--------|-------------|-------------|--------------------|
> | 0     | 0.0000 | 0.0007      | 0.0003      | 0.0000             |
> | 20    | 0.0122 | 0.0583      | 0.0569      | 0.0000             |
> | 40    | 0.2032 | 1.0248      | 0.9954      | 0.0000             |
> | 60    | 1.0055 | 4.7191      | 4.5890      | 0.0002             |
> | 80    | 2.0163 | 9.2662      | 9.0415      | 0.0003             |
> | 100   | 2.4094 | 11.0090     | 10.7649     | 0.0004             |
>
>     Table R1. Discrepancy learning by Equation (4)
>
> * **Optimizing Equation (6)**: We train the model by optimizing Equations (4) and (6) to learn the exact amount of discrepancy. Please note that the graph edit distance for the graph perturbed by two edges ($G_2$) is double the graph edit distance for the graph perturbed by only one edge ($G_1$). Here, we measure the representation distances of target nodes where the perturbation is applied. As shown in Table R2, the learning exact amount of discrepancy on global-level representations also **affects the local-level representations** resulting in the more distant node representations in more perturbed graphs (Node distance of ($G_0$, $G_2$) ) from the node representations in the original graph than node representations in less perturbed graphs (Node distance of ($G_0$, $G_1$) ).
> | Epoch | Graph Distance of ($G_0$, $G_1$) | Graph Distance of ($G_0$, $G_2$) | Node distance of ($G_0$, $G_1$)  | Node distance of ($G_0$, $G_2$) |
> |-------|------------|------------|-----------|-----------|
> | 0     | 0.0000     | 0.0001     | 0.0012    | 0.0008    |
> | 20    | 0.0002     | 0.0004     | 0.0013    | 0.0014    |
> | 40    | 0.0157     | 0.3019     | 0.0449    | 0.1358    |
> | 60    | 0.2199     | 0.4418     | 0.6241    | 1.8051    |
> | 80    | 0.9174     | 1.8228     | 2.6067    | 7.4082    |
> | 100   | 1.4597     | 2.9522     | 4.3504    | 12.1939   |
>
>     Table R2. Discrepancy learning by Equations (4) and (6)
>
> ---
>
> **Question 5.**  Why some perturbations in a large social network might drastically change some of its properties?
>
> **Answer 5.** The task that we deal with on social networks is a link prediction, and, in this link prediction task, perturbing a tiny amount of local information can drastically change its node representation, while the other nodes far away from this node might not be affected by this perturbation. In other words, regarding the local task, some perturbations on certain regions of nodes and edges can significantly affect the properties of that subregions more than the other regions.

---

> ### Author Response · Authors · 2022-08-01
> **Initial Response (3/5) to Reviewer NxMJ**
>
> **Question 4.** The proposed self-supervised loss in Equation 8 mainly focuses on the global information in a graph as Equations 4, 6, and 7 are all defined on the graph embedding level. This might cause the model not to pay attention to local information in the graph.
>
> **Answer 4.** This is a misunderstanding of our work since our D-SLA loss in Equation 8 is also **able to learn local information**. The objective of our D-SLA is to learn the local difference on the global graph-level representations and we achieved that objective by proposing a framework of learning the subtle (i.e., local) difference. we further clarify why our D-SLA can capture the local information.
>
> * At first, as represented in Equation (4), our D-SLA is trained by discriminating the original graph from its perturbed graphs, and there only are subtle differences between them when we perturb only the tiny amount of edges in the graph. Therefore, for optimizing the objective in Equation (4), the model should aware of the local differences between graphs.
> * Also, we propose to learn the exact amount of differences between original and perturbed graphs with the graph edit distance, formalized in Equation (6), and, to optimize the objective in Equation (6), our D-SLA should recognize how many edges are different between two graphs. That is, our D-SLA should focus on edge-level (local) differences, allowing it to capture the local information.
>
> Experimentally, we already showed our D-SLA does pay attention to the local information by evaluating it on the local-level task, namely link prediction, in Table 4 of Section 4.2, where ours significantly outperforms all the other graph SSL methods. Therefore, to summarize, as described in Lines 62-64, our D-SLA can not only capture local information (Section 4.2) but also discriminate global-level differences (Section 4.1) within/between graphs.
>
> Here, we provide explicit evidence that learning the local difference on graph-level representations can affect the local-level representations regarding Equations (4) and (6) respectively. Both experiments are conducted on a synthetic community graph with 4000 nodes and about 800K edges. We perturb only one or two edges to the original graph, making two perturbed graphs coined as $G_1$ and $G_2$. We compare the changes in graph-level representations with the changes in local-level representations. The metrics are as follows:
>
> 1) The distances between graph representations of the perturbed graphs and the original graph: $||h_{G_0} - h_{G_i}||$ where $h_{G_0}$ denotes the graph representation of original graph and $h_{G_i}$ denotes the graph representation of the $i$-th perturbed graph.
> 2) The distances between node representations of a node in the original graph and the corresponding node in the perturbed graph: $||h_{v_j,G_0} - h_{v_j,G_i}||$ where $h_{v_j,G_0}$ denotes the $j$-th node representation of the original graph and $h_{v_j,G_i}$ denotes the $j$-th node representation of the $i$-th perturbed graph.
>
> Due to the character limit, we provide the results in the comment box below.

---

> ### Author Response · Authors · 2022-08-01
> **Initial Response (2/5) to Reviewer NxMJ**
>
> **Question 2.**  How can we make sure that graph edit distance is a good proxy for measuring if two graphs should be close or far in the latent space?
>
> **Answer 2.** Graph edit distance is a good proxy since it is simple yet effective:
> 1) We can compute distances between any original and perturbed graphs in our perturbation method with **near-zero costs** (i.e., O(1)), as described in Lines 195-209 along with Table 1.
> 2) It is **generalizable to any graph domains**: we don’t have to control our perturbation and distance measures, even if we deal with graphs in unknown domains. In other words, if we use graph properties for measuring the distances among graphs, since properties are different across different graph domains, we might have to change our distance measures every time when we deal with a new graph domain. However, we don’t have to take this into consideration, thanks to graph edit distance.
> 3) It is **applicable to unlabeled datasets**: in graph SSL experiments, labels are generally not available during pre-training, and, with graph edit distance, we can compute distances between graphs without accessing any label.
>
> To summarize, we aim to make our D-SLA work in general regardless of domains (properties and labels of graphs), however, as described in Appendix C, it is also possible to pre-define semantic discrepancy to use in discrepancy learning, which we leave as future work.
>
> **Question 3.** The paper tested the proposed model for link prediction on social networks. However, I find the motivation of the paper to be weak in social networks (and stronger for small graphs like molecules).
>
> **Answer 3.**  This is a misunderstanding of our objectives in D-SLA. Our D-SLA aims to learn the **local difference**, which is the reason why we validate our D-SLA on social networks. Here, we clarify the motivation and the objective of our D-SLA.
> * The motivation for our D-SLA is that since graphs are discrete data structures, the property may largely vary even between slightly perturbed graphs as we described in Lines 40-42. Therefore, our D-SLA aims to learn the difference between slightly perturbed graphs and the original graph. As graph perturbation targets a local region, the property changes first occur in the subgraphical structures where the perturbation is applied. Then, the global properties are affected by the change in local semantics.
>
> * To this end, the objective of our D-SLA is to learn the local difference. However, if the model learns only the local-level representations, the model cannot learn the global graph-level representations as we tackled the drawbacks of predictive learning in Line 31. Therefore, the final objective of our D-SLA is to learn the local difference of the global graph-level representations.

---

> ### Author Response · Authors · 2022-08-01
> **Initial Response (1/5) to Reviewer NxMJ**
>
> We sincerely thank you for your constructive and helpful comments. We initially address all your concerns below:
>
> ---
>
> **Question 1.** The notion of distance here is a weighted version of the original contrastive learning: two similar graphs will still have small distances and be close in the latent space even if some properties of the perturbed graph drastically change from the original graph.
>
> **Answer 1.** This is a critical misunderstanding of our discrepancy learning framework, and our discrepancy learning clearly differs from contrastive learning. In particular, our D-SLA aims to learn the discrete embedding space by learning the discrepancy even between slightly perturbed graphs, thus obtaining a discriminative space for them that can further be utilized to distinguish between them for an unknown downstream task. Please note that graph self-supervised learning performs **unsupervised learning** of the graph representations with no knowledge of the downstream task, and the **properties** of graphs can drastically change from one downstream task to another.
>
> We first clarify what graph self-supervised learning aims to learn, and then describe the objectives of conventional graph contrastive learning. Finally, we clarify the difference between our D-SLA and conventional contrastive learning.
>
> * Self-supervised learning (SSL) aims to learn general information from the graph structure without utilizing any labels for downstream tasks. A self-supervised learner thus **cannot determine whether two similar graphs have similar or different properties**, since specific tasks and desired properties are not given in the pre-training stage with SSL. Learning a space that captures the **target properties** is something that is done at a **fine-tuning stage**, when a **specific downstream task** is given. As we demonstrated in Table 3, depending on the downstream tasks and finetuning datasets, some desired properties are related to the structural similarity such as ClinTox and BBBP, but some are not such as BACE and MUV. Thus a graph SSL should learn a space that can capture both **the similarity and difference between two graphs (i.e., accurate discrepancies among graphs)** without considering the labels.
>
> * Conventional graph contrastive learning methods aim to maximize the similarity between the representations of two similar graphs, overlooking the reality that even though they are similar, they could still have different properties. This pitfall results in the embedding collapse of similar graphs which could have different properties as shown in Figure 1 (b-2). Our method, on the other hand, learns discriminative space even for two graphs that are different only by a single node or an edge.
>
> * We devise a framework that can discretize graph embeddings according to their structures allowing the model can distinguish a subtle difference between similar graphs, whereas contrastive learning continuously maximizes the similarity between similar graphs. To solve the drawbacks of conventional contrastive learning, the model should capture not only the similarity between two similar graphs but also the subtle differences between them. To this end, we propose a discrepancy framework with completely opposite objective functions from contrastive learning as we described in Lines 67-69. Therefore, our objective functions are not similar to contrastive learning, but rather *completely opposite* from contrastive learning, since our objective functions aim to discriminate the similar graphs and learn the exact amount of discrepancy. That is, the small distance between two similar graphs as you mentioned is a key point of our D-SLA which allows learning the subtle difference between two similar graphs, which conventional contrastive learning cannot learn as shown in Figure 1 (b,c-2).
>
> We here clarify how learning the subtle difference affects the downstream tasks by recapitulating the experimental results.
>
> * As we demonstrated in Table 3 and Figure 4, the BACE dataset contains a bunch of **similar graphs with different properties**. Our D-SLA outperforms all baselines on this BACE dataset, whereas conventional contrastive learning methods do not outperform even predictive methods. Thus, as long as the graphs are well distinguished, even by small distances, we should allow the fine-tuning model to capture their differences. None of the existing graph SSL methods, either contrastive or predictive, can learn a discriminative space for slightly different graphs and thus obtain significantly lower performance compared to those obtained by our method.
>
> * In link prediction experiments in Section 4.2 where capturing local information is important, our D-SLA outperforms all baselines including conventional contrastive learning, which suggests that our D-SLA can capture the small difference in a local region by learning the subtle difference, whereas conventional contrastive learning methods fail to do so.

---

> ### Author Response · Authors · 2022-08-07
> **The end of the discussion phase is approaching**
>
> Dear Reviewer NxMJ
>
> We sincerely appreciate your positive comments on the importance of our tackled problems as well as our comprehensive experiments. During the response period, we have made every effort to faithfully address all your concerns/comments in the initial responses below. Here we briefly summarize the main points as follows:
> * We clarified that the objective of our discrepancy learning is **significantly different from conventional contrastive learning** and further demonstrated the differences between them by recapitulating the experimental results. For one particular example, **our objective is to learn the subtle difference** between two similar graphs on the graph-level representations, unlike contrastive learning which fails to do so.
> * We have clarified that optimizing our objective functions **allows the model to learn the local information** at the component level, and, to show the effectiveness of learning local information, we conducted experiments on the link prediction tasks. Also, we have further analyzed learned local information by our D-SLA **by measuring the discrepancy among local representations**.
> * We have further demonstrated the effectiveness of our hyperparameters regarding $\lambda_1$ and $\lambda_2$, by varying them.
>
> Since the end of the discussion phase is approaching, could you please go over our responses? Please let us know if you have anything else that we should address. We believe that including all the clarification and additional results in the response comments to our paper will significantly improve ours. And we thank you again for your time and efforts in reviewing our paper as well as your insightful and constructive comments.
>
> Best regards, Authors

---

### Official Review · Reviewer_3WcA · 2022-07-06

**Rating:** 4
**Confidence:** 4
**Soundness:** 2 fair
**Presentation:** 3 good
**Contribution:** 3 good

**Summary:**

This paper argues that existing graph self-supervised learning methods have some drawbacks. Specifically, predictive learning methods may not capture the global properties of graphs, and contrastive learning methods may not discriminate two similar graphs with different properties. To this end, a framework D-SLA is proposed to learn the exact discrepancy between the original and the perturbed graphs.

**Questions:**

Please respond to the weaknesses listed above.

**Ethics Review Area:**

["I don’t know"]

**Limitations:**

Yes. the authors have detailedly discussed the limitations and potential societal impacts in the appendix.

**Strengths And Weaknesses:**

- Strengths
>    1. The drawbacks of existing graph self-supervised methods are well analyzed and demonstrated.
>    2. The visualization experiment shows that the proposed method is quite effective.
>    3. The provided source codes facilitate the good reproducibility of this work.

- Weaknesses
>    1. There are also some existing works that do not rely on contrasting perturbed graphs, e.g., [1,2,3]. In this case, the argued drawbacks of existing methods claimed by the authors do not seem to hold.
>    2. Why use Equation (4) to discriminate original graphs from perturbed graphs? This could be explained more intuitively.
>    3. The proposed method D-SLA consists of many components and many hyperparameters. The sensitivity of the three key hyperparameters $\alpha$, $\lambda_1$, and $\lambda_2$ should be studied in the main text.
>    4. Most existing works report AUC and AP scores for the link prediction experiment. This work only reports accuracy.


*References*

[1] [arXiv 2022] Augmentation-Free Graph Contrastive Learning

[2] [SIGIR 2022] Are Graph Augmentations Necessary Simple Graph Contrastive Learning for Recommendation

[3] [WWW 2022] A Simple Framework for Graph Contrastive Learning without Data Augmentation

---

> ### Author Response · Authors · 2022-08-01
> **Initial Response (2/2) to Reviewer 3WcA**
>
> **Question 3.** The proposed method D-SLA consists of many components and many hyperparameters. The sensitivity of the three key hyperparameters $\alpha$, $\lambda_1$, and $\lambda_2$ should be studied in the main text.
>
> **Answer 3.** Thank you for the suggestion. Due to the page limit, we provide the sensitivity of $\lambda_1$ in Appendix B.4, with Figure S3 and Figure S4, where we observed that our D-SLA is not sensitive across different $\lambda_1$ values.
>
> On the other hand, based on your suggestion, we further provide the sensitivity of $\alpha$ and $\lambda_2$ by first varying their values during pre-training and then measuring the downstream performances on the BACE dataset. Note that, to measure the sensitivities of $\alpha$ and $\lambda_2$, we particularly chose the BACE dataset, since two structurally similar graphs are often different in their activities as shown in Table 3, thus learning subtle discrepancies is important for this dataset.
> | $\alpha$      | ROC-AUC        |
> |-------|----------------|
> | 1.0   | 83.75 ± 0.96 |
> | 5.0   | 83.81 ± 1.01 |
> | 10.0  | 78.34 ± 1.07 |
>
> Table R2. Effect of varying $\alpha$ on BACE dataset finetuning
>
> As shown in Table R2, when the margin $\alpha$ is too large, learning subtle discrepancies degenerates. The margin $\alpha$ in Equation (7) allows the model to preserve the discrepancy learned by Equations (4) and (6), since the model does not attract the embeddings of similar graphs if the embeddings of dissimilar graphs are sufficiently far apart than the margin $\alpha$ plus the distance between embeddings between similar graphs. We suggest that if the margin $\alpha$ is too large, the model tends to strongly attract the embeddings of similar graphs, resulting in the forgetting of the learned discrepancies between the similar graphs.
>
> | $\lambda_2$   | ROC-AUC        |
> |-----|----------------|
> | 0.1 | 83.68 ± 0.78 |
> | 0.5 | 83.81 ± 1.01 |
> | 0.9 | 80.72 ± 0.71 |
>
> Table R3. Effect of varying $\lambda_2$ on BACE dataset finetuning.
>
> As shown in Table R3, a larger $\lambda_2$  value degenerates the performance of our D-SLA, since too large $\lambda_2$  value forces the model to too focus on learning the discrepancy between two completely different graphs (i.e., perturbed and negative graphs), rather than learning the subtle differences among original and its slightly perturbed graphs.
>
> We will include these additional experimental results with varying hyperparameters in the revised version of the paper.
>
> ---
>
> **Question 4.** Most existing works report AUC and AP scores for the link prediction experiment. This work only reports accuracy.
>
> **Answer 4.** We sincerely appreciate your suggestion of using AUC and AP scores for link prediction. We have further evaluated baselines and our model with ROC-AUC and AP scores, and then reported the performances in Table R4. As shown in Table R4, **our D-SLA still clearly outperforms** all the other baselines on the link prediction task by large margins, even in AUC and AP.
> |  | ROC-AUC        | AP             |  ROC-AUC | AP |  ROC-AUC | AP
> |--------------|----------------|----------------|---|---|---|---|
> |  | **COLLAB** | | **IMDB-B** | |  **IMDB-M** | | |
> | No Pretrain  | 84.53 ± 0.55 | 80.01 ± 1.14 | 80.28 ± 2.23 | 68.72 ± 2.58 | 75.64 ± 1.42 | 64.93 ± 1.92 |
> | AttrMasking  | 85.07 ± 0.49 | 81.43 ± 0.80 | 81.78 ± 3.15 | 70.62 ± 3.68 | 74.26 ± 2.11 | 63.37 ± 2.15 |
> | ContextPred  | 86.49 ± 0.35 | 83.96 ± 0.75 | 80.49 ± 1.57 | 70.47 ± 2.24 | 74.20 ± 2.71 | 66.09 ± 2.74 |
> | Infomax      | 83.13 ± 0.35 | 80.83 ± 0.62 | 77.68 ± 1.70 | 67.25 ± 1.87 | 74.19 ± 1.85 | 64.98 ± 2.47 |
> | GraphCL      | 80.62 ± 0.88 | 76.04 ± 1.04 | 75.31 ± 3.07 | 63.71 ± 2.98 | 73.23 ± 3.16 | 62.40 ± 3.04 |
> | JOAO         | 81.58 ± 1.39 | 76.57 ± 1.54 | 76.80 ± 2.94 | 65.37 ± 3.23 | 73.72 ± 1.46 | 62.76 ± 1.52 |
> | GraphLoG     | 86.73 ± 0.65 | 82.95 ± 0.98 | 80.62 ± 2.29 | 69.71 ± 3.18 | 75.52 ± 1.82 | 64.88 ± 1.87 |
> | BGRL         | 81.56 ± 0.32 | 76.79 ± 1.13 | 79.18 ± 3.75 | 67.97 ± 4.14 | 74.74 ± 1.85 | 63.71 ± 2.09 |
> | **D-SLA (Ours)** | **88.14** ± 0.32 | **86.21** ± 0.38 | **86.64** ± 1.41 | **78.54** ± 2.79 | **78.53**  ± 1.51 | **69.45** ± 2.29|
>
> Table R4. Link prediction results with ROC-AUC and AP scores.
>
> We will include these new results in the revised version of the paper.

---

> ### Author Response · Authors · 2022-08-01
> **Initial Response (1/2) to Reviewer 3WcA**
>
> We sincerely thank you for your constructive and helpful comments. We appreciate your comments that our tackling drawbacks are well analyzed and demonstrated and our proposed method is effective in visualization experiments. We address all your concerns below:
>
> ---
>
> **Question 1.** There are also some existing works that do not rely on contrasting perturbed graphs, e.g., [1,2,3]. In this case, the argued drawbacks of existing methods claimed by the authors do not seem to hold.
>
> **Answer 1.**  Thank you for suggesting related works. The suggested AF-GCL[1] still has drawbacks, since different graphs are considered similar. To be more specific, for a graph in a batch, AF-GCL[1] chooses the most similar but structurally not the same graph in a batch and defines a graph and the most similar graph as a positive pair. Therefore, two graphs in a positive pair have different structures, although may have drastically different properties. As the objective function of AF-GCL is maximizing the similarity among structurally different graphs, AF-GCL could still suffer from the drawbacks of graph constrastive leanring approaches.
>
> SimGCL[2] and SimGRACE[3] augment the views of graphs by adding noise to model parameters or graph embeddings while preserving the graph structure. Therefore, their methods learn the similarity between structurally the same graphs. However, data augmentation is a key factor for self-supervised learning since, by data augmentation, the model can learn the representations of graphs not in the pretraining dataset and obtain more transferable representations. We further validate the significance of data augmentation by comparing graph classification performances.
> | Method       | BBBP           | ClinTox        | MUV            | HIV            | BACE           | SIDER          | Tox21          | ToxCast        | Avg.  |
> |--------------|----------------|----------------|----------------|----------------|----------------|----------------|----------------|----------------|-------|
> | SimGCL       | 67.37 ± 1.23 | 55.66 ± 4.72 | 71.24 ± 1.79 | 75.04 ± 0.86 | 74.11 ± 2.74 | 57.44 ± 1.74 | 74.39 ± 0.45 | 62.27 ± 0.38 | 67.19 |
> | SimGRACE     | 71.25 ± 0.86 | 64.16 ± 4.50 | 71.18 ± 3.40 | 74.52 ± 1.12 | 73.81 ± 1.37 | **60.59** ± 0.96 | 74.20 ± 0.64 | 63.36 ± 0.52 | 69.13 |
> | GraphCL       | 69.68 ± 0.67 | 75.99 ± 2.65 | 69.80 ± 2.66 | 78.47 ± 1.22 | 75.38 ± 1.44 | 60.53 ± 0.88 | 73.87 ± 0.66 | 62.40 ± 0.57 | 70.77 |
> | **D-SLA (Ours)** | **72.60** ± 0.79 | **80.17** ± 1.50 | **76.64** ± 0.91 | **78.59** ± 0.44 | **83.81** ± 1.01 | 60.22 ± 1.13 | **76.81** ± 0.52 | **67.24** ± 0.50 | **74.51** |
>
> Table R1. Finetuning results on graph classification with SimGCL and SimGRACE.
>
> As shown in Table R1, GraphCL which perturbs graph structure to augment views outperforms SimGCL and SimGRACE which preserve graph structures.This suggests that data augmentation allows the model to obtain more general and transferable representations.
> Additionally, **our D-SLA still outperforms the suggested works**, thanks to both data augmentation and discrepancy learning.
> We will include these results in our revised version of the paper.
>
> *References*
>
> [1] [arXiv 2022] Augmentation-Free Graph Contrastive Learning
>
> [2] [SIGIR 2022] Are Graph Augmentations Necessary Simple Graph Contrastive Learning for Recommendation
>
> [3] [WWW 2022] A Simple Framework for Graph Contrastive Learning without Data Augmentation
>
> ---
>
> **Question 2.** Why use Equation (4) to discriminate original graphs from perturbed graphs? This could be explained more intuitively.
>
> **Answer 2.** As described in Lines 166-167 with Figure 3 (b), the objective in Equation (4) embeds the perturbed graphs apart from the original graph. Then, by doing so, the model attempts to uniquely embed the original graph in the representation space, which leads the representation space to distinguish two similar graphs (e.g., the original graph and its slightly perturbed ones) having different properties, while learning their graph structures (i.e., node and edge features) as well.

---

> > ### Comment · Reviewer_3WcA · 2022-08-08
> > **This work is not well motivated.**
> >
> > Thank the authors for answering my questions. Regrettably, the authors did not resolve my major concern about the motivation of this work. In specific, the motivation of this work is based on the authors' argued limitation of contrastive learning, i.e., "*two differently perturbed graphs may result in representations that cannot discriminate two similar graphs with different properties*". However, as I have pointed out in Question 1, recently, there have been some recent advances in self-supervised learning (SSL) of GNNs, i.e., some recent works are able to effectively perform SSL on graphs without perturbing graphs. The given references [1,2,3] are such three example works. They have comprehensively justified that graph perturbations are not necessary for graph contrastive learning. They have also verified this for various tasks and on various widely-used datasets. Although the authors have experimentally compared studies [2,3] through the graph classification task, this is less convincing, because, in addition to the given references [1,2,3], we can reasonably believe that there are more recent works that do not have this limitation argued by the authors.
> >
> > Finally, I have also read the questions raised by the other reviewers. Reviewer NxMJ also raised concerns about the motivation of this work.
> >
> > Based on the above reasons, it seems that this work may require significant improvements, which cannot be completed during the camera-ready phase. Therefore, I would like to lower my rating from 5 to 4.

---

> > > ### Author Response · Authors · 2022-08-09
> > > **This is a critical misunderstanding, and we clarify our motivation again.**
> > >
> > > Q1. This work is not well motivated, since there exist works [1, 2, 3] that perform SSL on graphs without perturbing graphs.
> > >
> > > A1. This is a critical misunderstanding of our motivations. Please note that one of our main motivations is to **learn the exact discrepancies between different graphs**, and we use graph perturbation to create **slightly different graphs**. None of the existing works, including [1,2,3], aims to learn the exact discrepancy between two graphs, and **whether the baselines perturb the graphs or not does not affect our motivation, since none of them can learn how similar/dissimilar different graphs are, which ours aim to learn**.
> > >
> > > To go one step further regarding your misunderstanding, which we hope to be clarified in this response comment, “they [1, 2, 3] have comprehensively justified that graph perturbations are not necessary for graph contrastive learning”, yes we did point out the limitation of graph contrastive learning, but we also pointed out the limitation of these **baselines without perturbation**, in their inability to learn the exact difference between two graphs, in our paper as well as previous responses.
> > >
> > > We are sorry if our previous response leads you to a critical misunderstanding, but please see our Abstract in Lines 10-12 (and Introduction in Lines 67-69) that we aim to learn the exact discrepancy between graphs to discriminate two similar graphs with different properties. Once again, the suggested works [1, 2, 3] cannot discriminate between two similar graphs with the amounts of their exact distances, since they do not learn how similar the two graphs are, regardless of using the perturbations or not.
> > >
> > > During the rebuttal period, while we experimentally showed that our D-SLA significantly outperforms the suggested baselines in the graph classification task, you argued that graph classification tasks are not convincing enough to show the limitation – failure in exact discrepancy learning between graphs. However, the superior performances of our D-SLA come from its exact discrepancy learning, and that is, since baselines cannot learn the exact differences among graphs, they are suboptimal on downstream tasks. Furthermore, as shown in **our additional experiments on a link prediction task (See Table R5 in this response comment or Table S8 in the supplementary file)** our D-SLA also significantly outperforms baselines on this task as well due to its effectiveness in discrepancy learning.
> > >
> > > However, you might still be less convinced despite our two quantitative experiments above, and you might still have questions about 1) why exact discrepancy learning is helpful in downstream tasks, and 2) does suggested baselines really fail to learn such discrepancies between graphs. Regarding question 1), we already provided the evidence in our main paper in Table 3 and Table 4 that exact discrepancy learning is necessary for identifying similar graphs with different properties.
> > >
> > > Then, for the next question, **we visualize the learned embedding spaces with two different distance metrics, such as edit distance and Tanimoto similarity, in Figure S5 and Figure S6 of the supplementary file**, respectively. And, the embedding space shows that our D-SLA can capture the exact amount of discrepancy between different graphs, whereas **the suggested augmentation-free approaches cannot capture the topological differences of graphs**. Thus, to summarize, the whole discussions and their corresponding results in the above suggest that baselines have the limitation on learning the exact discrepancies, meanwhile, our D-SLA can capture them, which is one of our main motivations and which makes ours effective.
> > >
> > >
> > > |  | ROC-AUC        | AP             |  ROC-AUC | AP |  ROC-AUC | AP
> > > |--------------|----------------|----------------|---|---|---|---|
> > > |  | **COLLAB** | | **IMDB-B** | |  **IMDB-M** | |
> > > | SimGCL     | 81.56 ± 1.10 | 77.46 ± 0.86 | 76.29 ± 2.37 | 64.91 ± 2.60 | 74.60  ± 2.21 | 63.78 ± 2.28 |
> > > | SimGRACE        | 78.79 ± 1.07 | 74.51 ± 1.54 | 75.64 ± 2.40 | 64.49 ± 2.79 | 73.44 ± 2.15 | 62.81 ± 2.32 |
> > > | **D-SLA (Ours)** | **88.14** ± 0.32 | **86.21** ± 0.38 | **86.64** ± 1.41 | **78.54** ± 2.79 | **78.53**  ± 1.51 | **69.45** ± 2.29 |
> > >
> > > Table R5. Fine-tuning results on link prediction.
> > >
> > > ---
> > >
> > > Q2. Reviewer NxMJ also raised similar concerns.
> > >
> > > A2. Last but not least, you mentioned that Reviewer NxMJ is also concerned about our motivation similarly. However, the concern of Reviewer NxMJ is different from your concern since his/her main concern is that **the objective of our discrepancy learning might not work in social networks**. Regarding the answer to this question, please see our response to Reviewer NxMJ in Questions 3 and 4, in which we believe that we clearly deal with Reviewer NxMJ's concern.

---

> ### Author Response · Authors · 2022-08-07
> **The end of the discussion phase is approaching**
>
> Dear Reviewer 3WcA
>
> We sincerely appreciate your positive comments that we clearly motivate the drawbacks of existing graph SSL methods, we effectively visualize the strengths of our methods, and we provide our source code. During the response period, we have made every effort to faithfully address all your comments in the responses below. Here we briefly summarize the main points of our response as follows:
> * We have clarified that the suggested AF-GCL [1] still suffers from the drawback of contrastive learning. Then, we have validated that **the data augmentation is a key factor for obtaining transferable representations**, by showing that **our discrepancy learning outperforms suggested SimGCL [2] and SimGRACE [3]**.
> * We have clarified the effect of our graph discrimination task via the visualization of the embedding space, on which we have explained **the intuitive working principle of our graph discriminator**.
> * We have demonstrated the sensitivity of our hyperparameters regarding $\lambda_1$, $\alpha$, and $\lambda_2$, by varying them.
> * We have demonstrated that our D-SLA **outperforms** baselines **on ROC-AUC and AP scores** measures as well.
>
> Since the end of the discussion phase is approaching, could you please go over our responses? Please let us know if you have anything else that we should address. We thank you again for your time and efforts in reviewing our paper, and sincerely appreciate your insightful and constructive comments.
>
> Best regards, Authors
>
> ---
>
> *References*
>
> [1] [arXiv 2022] Augmentation-Free Graph Contrastive Learning
>
> [2] [SIGIR 2022] Are Graph Augmentations Necessary Simple Graph Contrastive Learning for Recommendation
>
> [3] [WWW 2022] A Simple Framework for Graph Contrastive Learning without Data Augmentation

---

### Official Review · Reviewer_yyyE · 2022-07-10

**Rating:** 7
**Confidence:** 3
**Soundness:** 3 good
**Presentation:** 3 good
**Contribution:** 3 good

**Summary:**

The paper presents a novel self-supervised learning approach on graphs to boost up the performance on downstream graph classification tasks. In many scenarios, two graphs with highly similar structure have remarkably different properties ((i.e. belong to different classes). By design, the classical contrastive learning-based approaches fail to separate out the representations of such similar graphs because they aim to maximize similarity between graphs and their perturbations. To address these limitations, the authors propose a novel approach referred to as Discrepancy-Based Self-Supervised Learning(D-SLA) that comprises of multiple pretext tasks including: i) training a discriminator to distinguish between a real graph and a perturbed variant; ii) learning the exact discrepancy between the original and the perturbed graphs, where the discrepancy is measured as the graph-edit distance. Additionally, increase the discrepancy measure between any two real graphs by introducing a margin factor, thereby enforcing that the original graph is embedded closer to its perturbed variants than any other  similar-looking real graph. The proposed approach outperforms the SOTA approaches from contrastive learning and
predictive learning literature on on several graph-related downstream tasks, including molecular property prediction, protein function prediction, and link prediction.

**Questions:**

1. For ablation study in Table 5, there needs to be another variant where graph-discriminator is turned off, but the discrepancy modeling is on. This will justify the additional utility of graph discriminator over the discrepancy-modeling pretext tasks.

**Limitations:**

I will encourage authors to discuss more on the possible intuition behind the efficacy of the proposed approach. To my understanding, discrepancy modeling with margin factor for real graphs is indirectly helping the model to identify key features (edges or nodes) that if perturbed, could lead to a valid graph with distinctive properties. Identification of such distinctive nodes/edges could largely aid to the downstream task of graph classification. Is that correct? If so, perhaps authors should clearly articulate this clearly in the paper as it might not be obvious to readers.

**Strengths And Weaknesses:**

Strengths:
1. The proposed approach is quite plausible and likely to be relevant to biological networks and molecular graphs where two similar graphs could have distinct properties.
2. The approach outperforms or matches the performance of several baselines from the fields of predictive and contrastive learning approaches on both graph classification and link prediction downstream tasks.
3. The paper is overall well-written and augmented with informative figures.


Weaknesses:
1. The approach involves a bunch of hyperparameters (e.g. the margin parameter \alpha, parameters \lambda_1 and \lambda_2 to determine the weight of each pretext task in the final objective function, max graph-edit distance considered to generate perturbed graphs). The paper does not include any guidelines on tuning these hyperparameters and how the choice of hyperparameters affect the results.

2. I  am not quite convinced if graph-discriminator pretext task is needed over L_{edit} and L_{margin}. (See my questions on table 5 below). It would be helpful if authors could throw insights on why all the three pretext tasks might be needed.

---

> ### Author Response · Authors · 2022-08-01
> **Initial Response (2/2) to Reviewer yyyE**
>
> **Question 3.** To my understanding, discrepancy modeling with margin factor for real graphs is indirectly helping the model identify key features (edges or nodes). Then, if real graphs are perturbed, key features could lead to a valid graph with distinctive properties. Consequently, the identification of such distinctive nodes/edges could largely aid the downstream task, for example, graph classification. Is that correct? If so, perhaps the authors should clearly articulate this clearly in the paper as it might not be obvious to readers.
>
> **Answer 3.** Thank you for your suggestion. We answer your statements one by one as follows:
>
> 1) Is discrepancy modeling with margin factor for real graphs indirectly helping the model identify key features (edges or nodes)?
>
>     Yes, learning the discrepancy between real graphs with a certain amount of margin can help the model accurately capture important nodes and edges, since the model should differentiate the different combinations of nodes and edges during discrepancy learning.
>
> 2) Then, if real graphs are perturbed, key features could lead to a valid graph with distinctive properties.
>
>     As you said, perturbed graphs might have distinctive representations which differ from their original graph representation throughout discrepancy learning. However, we cannot guarantee whether the perturbed graphs have valid graph structures, since, regarding molecular graphs, removing particular edges makes an invalid molecule structure. However, as described in Section C (i.e., Limitation and Potential Societal Impacts) of the supplementary file, it is infeasible to identify every valid structure and key features in a unified way for handling diverse domains, and we leave studying them as future work.
>
> 3) Consequently, does the identification of such distinctive nodes/edges could largely aid the downstream task, for example, graph classification?
>
>     Yes, even though the perturbed graphs might not be valid, we believe discriminative nodes/edges features from our discrepancy learning help achieve outstanding performances in downstream tasks.
>
> We will further clarify the above points in the revision.

---

> ### Author Response · Authors · 2022-08-01
> **Initial Response (1/2) to Reviewer yyyE**
>
> We sincerely thank you for your constructive and helpful comments. We deeply appreciate your comments that our proposed method is sound and the performance of our approach is meaningful compared to predictive and contrastive learning. We address all your concerns below:
>
> ---
>
> **Question 1.** The paper does not include any guidelines on tuning these hyperparameters (e.g. the margin parameter $\alpha$, parameters $\lambda_1$ and $\lambda_2$ to determine the weight of each pretext task in the final objective function,  max graph-edit distance considered to generate perturbed graphs) and how the choice of hyperparameters affect the results.
>
> **Answer 1.** Thank you for your helpful suggestion. We already provided the effect of $\lambda_1$ in Appendix B.4, with Figure S3 and Figure S4, where we observed that our D-SLA is not sensitive across different $\lambda_1$ values and the effect of perturbation magnitude in Appendix B.3, with Table 8, where we observed that weaker perturbation magnitude is helpful to capture the local semantics. Please note that the graph edit distance depends on the perturbation magnitude: a stronger perturbation magnitude causes a larger graph edit distance.
>
> On the other hand, based on your suggestion, we further provide the effects of $\alpha$ and $\lambda_2$ by first varying their values during pre-training and then measuring the downstream performances on the BACE dataset. Note that, to measure the sensitivities of $\alpha$ and $\lambda_2$, we particularly chose the BACE dataset, since two structurally similar graphs are often different in their activities as shown in Table 3, thus learning subtle discrepancies is important for this dataset.
>
> | $\alpha$      | ROC-AUC        |
> |-------|----------------|
> | 1.0   | 83.75 ± 0.96 |
> | 5.0   | 83.81 ± 1.01 |
> | 10.0  | 78.34 ± 1.07 |
>
> Table R1. Effect of varying $\alpha$ on BACE dataset finetuning
>
> As shown in Table R1, when the margin $\alpha$ is too large, learning subtle discrepancies degenerates. The margin $\alpha$ in Equation (7) allows the model to preserve the discrepancy learned by Equations (4) and (6), since the model does not attract the embeddings of similar graphs if the embeddings of dissimilar graphs are sufficiently far apart than the margin $\alpha$ plus the distance between embeddings between similar graphs. We suggest that if the margin $\alpha$ is too large, the model tends to strongly attract the embeddings of similar graphs, resulting in the degeneration of learning discrepancies between the similar graphs.
>
> | $\lambda_2$   | ROC-AUC        |
> |-----|----------------|
> | 0.1 | 83.68 ± 0.78 |
> | 0.5 | 83.81 ± 1.01 |
> | 0.9 | 80.72 ± 0.71 |
>
> Table R2. Effect of varying $\lambda_2$ on BACE dataset finetuning
>
> As shown in Table R2, a larger $\lambda_2$  value degenerates the performance of our D-SLA, since too large $\lambda_2$  value forces the model to too focus on learning the discrepancy between two completely different graphs (i.e., perturbed and negative graphs), rather than learning the subtle differences among original and its slightly perturbed graphs.
>
> We further add the effect of varying our hyperparameters in the next revision.
>
> ---
>
> **Question 2.** I am not quite convinced if graph-discriminator pretext task is needed over $L_{edit}$ and $L_{margin}$. For ablation study in Table 5, there needs to be another variant where graph-discriminator is turned off, but the discrepancy modeling is on.
>
> **Answer 2.** We already provided the ablation study that we only use the edit distance loss (i.e., $L_{edit}$) without the graph discriminator loss (i.e., $L_{GD}$) in Table 5. As shown in the table, if we only use the edit distance loss, the performance of our D-SLA becomes degenerate, since the model would trivially set all the distances between the original and its perturbed graphs as zero which is discussed in Lines 221-226 and Lines 347-350. Thus, joint training of our graph discriminator loss (i.e., $L_{GD}$) and edit distance loss (i.e., $L_{edit}$) is a necessity.

---

> ### Author Response · Authors · 2022-08-07
> **The end of the discussion phase is approaching**
>
> Dear Reviewer yyyE
>
> We sincerely appreciate your positive comments on the soundness of our proposed method, the clarity of our paper, and the significance of the experimental results. We have made every effort to faithfully address all your comments in the responses, and here we briefly summarize the main points of our response as follows:
> * We have demonstrated the sensitivity of our hyperparameters by varying them and then provided the guidelines for the choice of hyperparameters.
> * We have clarified **the necessity of joint training** of the graph discrimination and exact discrepancy learning tasks.
>
> Since the end of the discussion phase is approaching, could you please go over our responses? Please let us know if you have anything else that we should address. We thank you again for your time and efforts in reviewing our paper as well as your constructive comments.
>
> Best regards, Authors

---

### Official Review · Reviewer_BmqR · 2022-07-11

**Rating:** 5
**Confidence:** 4
**Soundness:** 3 good
**Presentation:** 3 good
**Contribution:** 2 fair

**Summary:**

This paper proposes a discrepancy-based framework to learn graph representations. The framework leverages the discrepancy between 1) the original graph and the perturbed graphs and 2) the original graph with other negative graphs. The discrepancy amount for perturbed graph is defined via graph edit distance, which is easy to obtain by perturbed graph generating process. The discrepancy amount for other negative graphs are defined above a margin. Under such discrepancy hierarchy, the learned graph representations could preserve dissimilarity among graphs accurately. The learned representations could be used for downstream tasks such as graph classification and link predictions in chemical and biological domains or social networks.

**Questions:**

1. This method has data augmented by generating perturbed graph via eq(5). I wonder is this a common approach for obtaining perturbed graph, or a novel part of this paper? For those contrastive learning method papers which also consider original graph and perturbed graphs, are the perturbed graphs being generated, or being found by calculating graph edit distance?

2. This approach first generates perturbed graphs and defines discrepancy as graph edit distance. Then all the other samples are viewed as negative samples, and eq(7) assumes distance between original graph and negative graph is greater than a margin. This assumes that all the other graphs are very different from the original graph, at least not as some perturbation of original graph. I wonder could it be the case that in real data the graph samples already contains graphs similar to the original graph. For example, in Figure 1, (d) and (e) are two similar graphs in the samples, and the edit distance between (d) and (e) appear to be 1. Based on the proposed method, it will first generates some perturbed graph of (d), and then define distance of (d) and (e) greater than $\alpha$. In this example, distance(d,e)  > distance(d, perturbed graph of d), is this still a reasonable assumption? How the graph sample (e) different from those generated perturbation of (d)?

In summary, my question is: how the method differentiate the generated perturbed graphs, and the graphs that are similar to the original graphs in the sample? Will the assumption that all the other graphs are negative samples and have distance at least $\alpha$ actually disturb learning and makes the learning results worse, when they are actually similar to the original graph? Consider the extreme cases that the graph samples are all perturbations of one graph, what graph representations the proposed method would give?

3. How to choose the number of perturbed graphs n, and if the perturbed graphs samples are large, would the imbalance of two classes (1 vs n) affect the performance of classifier/discriminator?


**Ethics Review Area:**

["I don’t know"]

**Limitations:**

The authors have discussed limitations thoroughly in the supplementary materials. I do not have other to add except the ones listed in Questions and Strengths And Weaknesses sections.

**Strengths And Weaknesses:**

Originality
This paper proposes a new discrepancy preserved framework for learning graph representations, addressing limitations in existing literatures that would learn similar representations of two perturbed graphs with different properties. However, I would like to understand the novelty of the framework as how the discrepancy framework is different from redefining the similarity functions, e.g., sim(two perturbed graphs) = graph edit distance, and sim(graph, negative graph) = $\alpha$ + additional terms, and then use existing methods. Would appreciate if the authors could further clarify it.

Quality:
The paper is well written. The experiments are overall comprehensive, and additional analysis support the main points well.

Clarity:
The methodologies and implementation details are clearly explained in the main article and supplementary material.

Significance
As mentioned in the originality section, this paper uses discrepancy preserved framework to learn graph representations. However, my understanding is that authors define a new similarity function, which leverages the discrepancy among graphs. The significance is not quite obvious, and would appreciate authors' further clarifications.

---

> ### Author Response · Authors · 2022-08-01
> **Initial Response (3/3) to Reviewer BmqR**
>
> **Question 4.** I wonder could it be the case that in real data the graph samples already contains graphs similar to the original graph.
>
> **Answer 4.** There could be more similar negative graphs than perturbed ones, however, such a situation rarely happens, since the pre-training dataset contains general graphs collected from the real world and our perturbation method makes only a subtle difference from the original one.
>
> ---
>
> **Question 5.** How to choose the number of perturbed graphs n, and if the perturbed graphs samples are large, would the imbalance of two classes (1 vs n) affect the performance of classifier/discriminator?
>
> **Answer 5.** Our discriminator does not suffer from the class imbalance when the number of perturbed graphs is large since we design our graph discrimination to increase the score of the original graphs *compared* to the scores of the perturbed graphs. Specifically, our D-SLA does not compute a loss value per score or per graph, as the independent computation of loss for each graph could cause the class imbalance problem. Instead, our D-SLA computes one loss value by combining the scores of the original and perturbed graphs and aims to maximize the score of the original graph compared to the scores of the perturbed graphs.
>
> On the other hand, if the number of perturbed graphs is large, the model can learn the more diverse graph structures and may obtain more transferable representations for various downstream tasks. However, we argue that the number of perturbed graphs has a trade-off relationship with time and memory space for perturbing more graphs.

---

> ### Author Response · Authors · 2022-08-01
> **Initial Response (2/3) to Reviewer BmqR**
>
> **Question 2.** I wonder is this a common approach for obtaining perturbed graph, or a novel part of this paper?
>
> **Answer 2.** Inspired by our graph discriminator, we propose a perturbation method that is challenging for the model to discriminate by learning the subtle difference between the perturbed graphs and the original graph. Please note that the perturbation strategy follows along with the self-supervised learning strategy.
>
> Conventional contrastive learning [1,2] considers that graph perturbation does not change the local and global properties. Consequently, over 20% of nodes or edges are perturbed in contrastive learning methods by attribute masking, edge perturbation, node dropping, or subgraph sampling.
>
> Contrarily, we argue that perturbing even a subtle region can change the local and global properties and the model should distinguish the subtle differences between the perturbed graphs and the original graph. Therefore, our perturbation strategy is different from the conventional perturbation strategy since **we aim to perturb only a subtle region** of a graph and make the graph discriminator confuse to discriminate the original graph from the perturbed graphs. Specifically, we clarify which component is different from conventional perturbation strategies:
> 1) We choose to perturb **edges**, since perturbing nodes (i.e., node dropping and subgraph sampling) significantly changes the graph properties.
> 2) We perturb only a **small amount of edges** by aiming to discriminate the subtle differences between the perturbed graphs and the original graph.
> 3) In our framework, attribute masking is viewed as an auxiliary device to **make it harder to distinguish the subtle difference**. In our point of view, the attribute masking allows the model to more focus on learning the subtle differences in connectivity.
>
> In our D-SLA framework, attribute masking plays an important role to learn the subtle difference, since it is easy to distinguish two similar graphs when giving the all information about nodes. To validate this fact, We further demonstrate the effect of attribute masking for capturing the local semantics on the link prediction task.
> |    COLLAB   | Accuracy       |
> |-------------|----------------|
> | w/ Masking  | 76.19 +/- 0.50 |
> | w/o Masking | 70.42 +/- 0.95 |
>
> Table R1. Ablation study on attribute masking.
>
> As shown in table R1, the performance without attribute masking is significantly lower than the performance with attribute masking. We suggest that, in the pre-training stage, attribute masking actually limits the information given to the model and forces the model to learn more transferable and fruitful representations, demonstrating that attribute masking in our perturbation is a key factor to learn the local semantics.
>
> *Reference*
>
> [1] You, Yuning, et al. "Graph contrastive learning with augmentations." Advances in Neural Information Processing Systems 33 (2020): 5812-5823.
>
> [2] You, Yuning, et al. "Graph contrastive learning automated." International Conference on Machine Learning. PMLR, 2021.
>
> ---
>
> **Question 3.** For those contrastive learning method papers which also consider original graph and perturbed graphs, are the perturbed graphs being generated, or being found by calculating graph edit distance?
>
> **Answer 3.** The conventional contrastive learning methods generate the perturbed graphs by selecting a specific perturbation strategy without leveraging graph edit distance.

---

> ### Author Response · Authors · 2022-08-01
> **Initial Response (1/3) to Reviewer BmqR**
>
> We sincerely thank you for the constructive and helpful comments. We appreciate your comments that the methodologies and implementations of our proposed method are clearly written and experimental results including analysis well support our argument. We address all your concerns below:
>
> ---
>
> **Question 1.** I would like to understand the novelty of the framework as how the discrepancy framework is different from redefining the similarity functions, and then use existing methods. The significance is not quite obvious, and would appreciate authors' further clarifications.
>
> **Answer 1.** The significance of our work is **tackling a fundamental assumption** of conventional graph contrastive learning approaches which overlooks the fact that graphs are discrete structures, and we proposed a novel framework with a **completely opposite objective** from conventional contrastive learning to tackle the problem.
>
> Graph contrastive learning approaches assume that the perturbed graphs are similar to the original one. However, we argue that such a fundamental assumption does not hold in the graph domain, observing that similar graphs could have largley different properties.
>
> To this end, we do not redefine the similarity, but rather propose completely opposite objectives from the conventional contrastive learning schemes as described in line 67-69.
>
> Specifically, contrasting to the conventional contrastive learning methods, our graph discriminator treats the **perturbed graphs as dissimilar** and enforces the model to **embed them apart** in the latent space (Figure 3 (b)). Also, we propose a method to learn the exact amount of discrepancy between the perturbed graphs and the original graph, whereas conventional contrastive learning try to maximize the similarity between them.
>
> By tackling that the fundamental assumption in conventional contrastive learning does not hold in the graph domain, we believe that our discrepancy learning framework could give new insight into the graph self-supervised field.
>
> Additionally, we proposed an **efficient way to measure the discrepancy** by leveraging the **graph edit distance** which is computed at **near-zero cost** while perturbing graphs.

---

> ### Author Response · Authors · 2022-08-07
> **The end of the discussion phase is approaching**
>
> Dear Reviewer BmqR
>
> We sincerely appreciate your positive comments in regard to the quality and clarity of our paper. During the response period, we have made every effort to faithfully address all your concerns/comments, given in detail below. In short,
> * We have clarified the significance of our work in two folds. At first, **we tackle a fundamental problem of conventional graph contrastive learning** which has been overlooked in the graph domain. Also, we have clarified **the difference between conventional contrastive learning and our discrepancy learning** by comparing each component of our discrepancy learning against conventional contrastive learning.
> * We have explained the objective of our perturbation strategy and then clarified **the difference in perturbation strategies** between ours and conventional contrastive learning.
> * We have clarified that our graph discriminator is not affected by the class imbalance problem.
>
> Since the end of the discussion phase is approaching, could you please go over our responses? Please let us know if you have anything else that we should address. We thank you again for your time and efforts in reviewing our paper, and sincerely appreciate your insightful comments.
>
> Best regards, Authors

---

### Author Response · Authors · 2022-08-02
**Recap of our main novelty and contributions**

We sincerely appreciate your time and effort in reviewing our papers, as well as the constructive comments and valuable suggestions. Here, we clarify our novelty and contribution in the following two aspects: **framework-level** and **component-level**. For the other points you asked or raised beside the main contributions, please refer to our responses to each question in the comments for each reviewer.

---

## Framework-level novelty
* We first want to emphasize that our main contribution is the **learning of subtle differences between two similar graphs**, which has been overlooked in conventional graph contrastive learning. In other words, we aim to learn a discriminative representation space, since graphs are discrete data structures and two slightly different graphs may have drastically different properties. We find that learning the subtle difference between two similar graphs is significant to obtain transferable representations as shown in Tables 2 and 4. We strongly believe that our framework cannot be treated as similar to conventional contrastive learning, but rather be significantly different.
* Our discrepancy learning framework aims to learn the local difference **on graph-level representations**, by tackling that predictive learning cannot learn the global representations. We find that our proposed framework can **capture not only the local semantics but also the global semantics** as we demonstrated in Tables 2 and 4, and Figure 5.
* Our discrepancy learning framework is a **general** graph self-supervised learning method that is applicable to diverse domains not limited to molecular graphs, but also to biological and social networks since our proposed framework can capture both local and global information.

---

## Component-level novelty
We now highlight the difference between our components and contrastive learning.
* Our graph discrimination task in Section 3.2 forces the model to **embed two similar graphs into distinct representations** (Figure 3 (b)), which is completely opposite from the objective of contrastive learning.
* Our discrepancy learning with graph edit distance in Section 3.3 allows the model to **learn the discrete embedding space** according to discrete graph structure even if two graphs are highly similar (Figure 3 (c)), whereas contrastive learning continuously attracts the embeddings of similar graphs and cannot learn the discrete embedding space.
* We facilitate learning the exact amount of discrepancy by leveraging graph edit distance which is computed with **near-zero cost** when performing graph perturbations.

---

### Meta-Review · Area_Chair_WuE2 · 2022-08-29

**Recommendation:** Accept
**Confidence:** Less certain

**Metareview:**

This paper proposes a novel self-supervised learning strategy by considering the quantitative discrepancy of two perturbed graphs, which is measured by graph edit distance. The major concerns come from the motivation of the proposed approach. This has been well addressed in authors’ rebuttal, with additional new experiments. The authors have done a great job in addressing this main concern and other questions raised by reviewers, such as ablation studies on major hyperparameters. The contribution of incorporating graph-level quantitative metric as additional self-supervision signal is clear. Although there are divided ratings in the end, I still recommend acceptance of this paper.

**Award:**

No

---

### Decision · Program_Chairs · 2022-09-14

Accept